# High-dimensional Bayesian Optimization via Covariance Matrix Adaptation Strategy

**Lam Ngo**                                                        *s3962378@student.rmit.edu.au*
*RMIT University, Australia*

**Huong Ha**                                                        *huong.ha@rmit.edu.au*
*RMIT University, Australia*

**Jeffrey Chan**                                                    *jeffrey.chan@rmit.edu.au*
*RMIT University, Australia*

**Vu Nguyen**                                                       *vutngn@amazon.com*
*Amazon, Australia*

**Hongyu Zhang**                                                    *hyzhang@cqu.edu.cn*
*Chongqing University, China*

**Reviewed on OpenReview:** *https://openreview.net/forum?id=eTgxr7gPuU*

## Abstract

Bayesian Optimization (BO) is an effective method for finding the global optimum of expensive black-box functions. However, it is well known that applying BO to high-dimensional optimization problems is challenging. To address this issue, a promising solution is to use a local search strategy that partitions the search domain into local regions with high likelihood of containing the global optimum, and then use BO to optimize the objective function within these regions. In this paper, we propose a novel technique for defining the local regions using the Covariance Matrix Adaptation (CMA) strategy. Specifically, we use CMA to learn a search distribution that can estimate the probabilities of data points being the global optimum of the objective function. Based on this search distribution, we then define the local regions consisting of data points with high probabilities of being the global optimum. Our approach serves as a *meta-algorithm* as it can incorporate existing black-box BO optimizers, such as `BO`, `TuRBO` (Eriksson et al., 2019), and `BAxUS` (Papenmeier et al., 2022), to find the global optimum of the objective function within our derived local regions. We evaluate our proposed method on various benchmark synthetic and real-world problems. The results demonstrate that our method outperforms existing state-of-the-art techniques.

## 1 Introduction

Optimizing expensive black-box functions is an important task that has various applications in machine learning, data science, and operational research. Bayesian Optimization (BO) (Jones et al., 1998; Brochu et al., 2010; Shahriari et al., 2016; Binois & Wycoff, 2022; Garnett, 2023) is a powerful approach to tackle this challenging problem in an efficient manner. It has been successfully applied in a wide range of applications, including but not limited to hyperparameter tuning of machine learning models (Snoek et al., 2012; Turner et al., 2020), neural architecture search (Jenatton et al., 2017; Kandasamy et al., 2018b), material design (Hernández-Lobato et al., 2017), robotics (Calandra et al., 2016), and reinforcement learning (Brochu et al., 2010; Parker-Holder et al., 2022).

BO operates in an iterative fashion by repeatedly training a *surrogate model* based on the observed data and using an *acquisition function* to suggest promising data points for evaluation (Garnett, 2023). This method is inspired by Bayes' theorem, which aims to improve the prior belief about the objective function by incorporating observed data to obtain a posterior with better information. In this way, BO selects the next data points by considering previous information and maximizes the knowledge gained about the objective function with new observations, making it sample-efficient in finding the objective function's global optimum.

Despite being a powerful optimization method, BO still suffers from various problems, including the curse of dimensionality issue, i.e., it often performs poorly when applied to high-dimensional problems (Rana et al., 2017; Eriksson et al., 2019; Binois & Wycoff, 2022; Papenmeier et al., 2022). One reason for this is that, as the search domain grows in dimensionality, more local optima may appear, making it difficult for the algorithm to find the global optimum. Additionally, a larger search space also implies more regions with high uncertainty, which can potentially cause the acquisition function to overemphasize exploration and fail to exploit potential regions within a fixed budget. Moreover, in high-dimensional spaces, the objective function is typically heterogeneous, making it difficult to fit a global surrogate model across the entire domain.

There have been various works attempting to make BO work well for high-dimensional optimization problems. One of the most promising approaches, which have demonstrated significant success, is the use of a local search strategy that partitions the search domain into promising local regions where the optimization process is performed within (Munos, 2011; Wang et al., 2014; Eriksson et al., 2019; Wang et al., 2020; Wan et al., 2021). These works, however, have certain limitations. For instance, the works in Munos (2011); Wang et al. (2014); Eriksson et al. (2019) employ a search space partition technique with fixed parameters that maybe difficult to optimally specify in advance, and thus, may not provide adequate flexibility for various problems (Wang et al., 2020). The work in Wang et al. (2020) learns promising local regions by partitioning the search space into non-linear boundary regions using an unsupervised classification algorithm, but there is no guarantee that these local regions can be learned accurately with a limited amount of training data.

In this paper, we follow the aforementioned local search approach to tackle the high-dimensional optimization problem. We propose to use the Covariance Matrix Adaptation (CMA) strategy to systematically define the local regions to be used within this local search approach. CMA is a technique developed in the Evolutionary Algorithm literature (Hansen & Ostermeier, 2001), aiming to estimate the probability distribution of data points in the search domain being the global optimum of the objective function (i.e., *search distribution*). Typically, CMA is combined with Evolutionary Strategy (ES) techniques like `CMA-ES` (Hansen & Ostermeier, 2001). These techniques leverage the search distribution derived from CMA to guide the search for the global optimum toward promising regions in the search domain, i.e., regions that highly likely contain the global optimum. It has been shown that CMA-based ES techniques, such as `CMA-ES`, perform very well in finding the global optima of high-dimensional optimization problems, demonstrating CMA's effectiveness in identifying promising regions that likely contain the global optima of high-dimensional optimization problems. A drawback of these techniques is that they generally require a large number of function evaluations (Loshchilov & Hutter, 2016; Nomura et al., 2021).

Inspired by the effectiveness of CMA in working with high-dimensional optimization problems, we propose to incorporate CMA into BO methods by using it to define the local regions in the BO local search approach. In particular, we define the local regions as the regions with the highest probabilities of containing the global optimum based on CMA's search distribution. Subsequently, we can use an existing BO optimizer, e.g., `BO`, `TuRBO` (Eriksson et al., 2019), `BAxUS` (Papenmeier et al., 2022), within these local regions to find the global optimum of the objective function. By leveraging information from CMA's search distribution, BO methods can focus the search within the local regions that have high likelihood of containing the global optimum of the optimization problem. CMA-based BO methods are therefore expected to work well with high-dimensional optimization problems, due to CMA's effectiveness, whilst preserving data-efficiency, a property lacking in CMA-based ES techniques. We derive the `CMA-BO`, `CMA-TuRBO` and `CMA-BAxUS` algorithms corresponding to the cases when we incorporate CMA with the optimizers `BO`, `TuRBO`, `BAxUS`, respectively. Our experimental results on various synthetic and real-world benchmark problems confirm that our proposed approach helps existing BO methods to work better for high-dimensional optimization problems whilst being data-efficient.

In summary, our contributions are as follows:

- Proposing a novel *meta-algorithm* using the local search approach and the CMA strategy to enhance the performance of existing BO methods for high-dimensional optimization problems;

- Deriving the CMA-based BO algorithms corresponding to the cases when we incorporate the proposed meta-algorithm with the state-of-the-art BO methods (e.g., `BO`, `TuRBO`, `BAxUS`);

- Conducting a comprehensive evaluation on various high-dimensional synthetic and real-world benchmark problems and demonstrating that our proposed CMA-based meta-algorithm outperforms existing state-of-the-art methods.

The implementation of our method is available at `https://github.com/LamNgo1/cma-meta-algorithm`.

## 2 Background

In this section, we present the fundamental background of BO. Then we revisit two state-of-the-art BO methods (`TuRBO`, `BAxUS`) that we will incorporate into our CMA-based meta-algorithm.

### 2.1 Bayesian Optimization

Bayesian optimization (BO) is a powerful optimization method to find the global optimum of an expensive black-box objective function by sequential queries (Jones et al., 1998; Brochu et al., 2010; Shahriari et al., 2016; Husain et al., 2023; Garnett, 2023). Let us consider the minimization problem: given an unknown objective function $f : \mathcal{X} \to \mathbb{R}$ where $\mathcal{X} \subset \mathbb{R}^d$ is a compact space, the goal of BO is to find a global optimum $\boldsymbol{x}^*$ of the objective function $f$:

$$\boldsymbol{x}^* \in \arg\min_{\boldsymbol{x} \in \mathcal{X}} f(\boldsymbol{x}). \tag{1}$$

BO solves an optimization problem in an iterative manner. First, the objective function $f$ is approximated by a *surrogate model* trained with the current observed dataset $D_0 = \{\boldsymbol{x}_i, y_i\}_{i=1}^{t_0}$ with $y_i = f(\boldsymbol{x}_i) + \varepsilon_i$ and $\varepsilon_i \sim \mathcal{N}(0, \sigma^2)$ being the corrupted noise. Then an *acquisition function* $\alpha : \mathcal{X} \to \mathbb{R}$ is constructed from the surrogate model to assign scores to all data points in the domain $\mathcal{X}$ based on their potential to improve the optimization process. The next data point to be evaluated, denoted as $\boldsymbol{x}_{\text{next}}$, is selected as the maximizer of the acquisition function. Subsequently, the objective function $f$ is evaluated at $\boldsymbol{x}_{\text{next}}$ and the new observed data $(\boldsymbol{x}_{\text{next}}, y_{\text{next}})$, with $y_{\text{next}} = f(\boldsymbol{x}_{\text{next}}) + \varepsilon$ and $\varepsilon \sim \mathcal{N}(0, \sigma^2)$, is added to the current observed dataset. The process is conducted iteratively until a pre-defined budget is exhausted, and then BO returns the best value found from the observed dataset as an estimate of the global optimum $\boldsymbol{x}^*$.

There are different choices for the surrogate models to be used in BO, including Gaussian Process (GP) (Rasmussen & Williams, 2006), Tree-structured Parzen Estimator (TPE) (Bergstra et al., 2011), and neural networks (Springenberg et al., 2016). In our work, we focus on the GP surrogate model, which is one of the most popular surrogate models in BO. GP is a probabilistic model that can provide both scalar predictions for the objective function values and their associated uncertainty. A GP is completely specified by its mean function $\mu(\boldsymbol{x})$ and covariance function (kernel) $k(\boldsymbol{x}, \boldsymbol{x}')$ (Rasmussen & Williams, 2006). While the mean function indicates the most probable values for the objective function values, the covariance function captures the properties of the objective function, e.g. its smoothness.

There is also a variety of common acquisition functions. Examples include Expected Improvement (EI) (Mockus et al., 1978), Probability of Improvement (PI) (Kushner, 1964), Upper Confidence Bound (UCB) (Srinivas et al., 2010), Thompson Sampling (TS) (Thompson, 1933), and Knowledge Gradient (KG) (Frazier et al., 2009). While each of these acquisition functions has its own strengths and weaknesses, they all aim to balance between exploration and exploitation. Exploration encourages the algorithm to look for promising values in highly uncertain locations, whilst exploitation favors refining the knowledge around the currently optimal locations. In this work, we use Thompson Sampling (TS) (Thompson, 1933) as the acquisition function following Eriksson et al. (2019); Papenmeier et al. (2022). In the next section, we will describe in detail the TS acquisition function which will be later used in our method.

## 2.2 The Thompson Sampling Acquisition Function

The Thompson Sampling (TS) acquisition function (Thompson, 1933), commonly used in BO research (Kandasamy et al., 2018a; Eriksson et al., 2019; Papenmeier et al., 2022), follows a stochastic policy. The main idea is to sample a random realization (i.e., *sample path*) of the objective function from its posterior distribution, then optimize this sample path to find the next data point to be evaluated. Specifically, at iteration $t$, given the observed dataset $D_t$, the TS acquisition function $\alpha^{\mathrm{TS}}(\boldsymbol{x}; D_t)$ can be defined as,

$$\alpha^{\mathrm{TS}}(\boldsymbol{x}; D_t) = f^{(t)}(\boldsymbol{x}) \quad \text{where } f^{(t)}(\boldsymbol{x}) \sim p(f|D_t), \tag{2}$$

and $p(f|D_t)$ denotes the posterior distribution of the objective function $f$ given the observed data $D_t$. If GP is used as the surrogate model for the objective function, then $f^{(t)}(\boldsymbol{x}) \sim \mathrm{GP}(\mu(\boldsymbol{x}), k(\boldsymbol{x}, \boldsymbol{x}')|D_t)$.

For a minimization problem as defined in Eq. (1), the TS process is then to minimize the acquisition function $\alpha^{\mathrm{TS}}(\boldsymbol{x}; D_t)$ to find the next data point $\boldsymbol{x}_{t+1}$ to be evaluated,

$$\boldsymbol{x}_{t+1} = \arg\min_{\boldsymbol{x} \in \mathcal{X}} \alpha^{\mathrm{TS}}(\boldsymbol{x}; D_t). \tag{3}$$

The TS acquisition function selects the data point to be evaluated by drawing a random realization of the objective function $f$ from its posterior distribution, therefore, it encourages exploitation of regions with higher probabilities of being optimal, while allowing for exploration of other regions, owing to its random nature. It thus satisfies the desirable property of balancing between exploitation and exploration, which is a requirement for any acquisition function in BO.

## 2.3 TurBO

`TuRBO` (Eriksson et al., 2019) is a state-of-the-art BO method, which proposes to use the local search strategy to solve the high-dimensional optimization problem. In `TuRBO`, the global search space is partitioned into smaller domains, called trust regions (TR) (Yuan, 2000), within which the surrogate model is believed to accurately model the objective function. `TuRBO` uses GP as the surrogate model and trains the GP using all the previous observed data then selects the next observed data point by optimizing the acquisition function locally within the TR. This local strategy makes `TuRBO` more efficient than standard `BO` methods in high-dimensional problems, as it only requires modeling of local surrogate models, abandoning the global surrogate model used in standard `BO` methods. The local surrogate models of `TuRBO` do not suffer from the heterogeneity of the objective function, as they only need to accurately capture the objective function within the TR. Moreover, as the TR reduces the regions with large uncertainty, `TuRBO` mitigates the over-exploration issue commonly encountered by standard `BO` methods when solving high-dimensional optimization problems. The detailed description of `TuRBO` can be found in Appendix Section A.1.

## 2.4 BAxUS

`BAxUS` (Papenmeier et al., 2022) is a state-of-the-art BO method that tackles the high-dimensional optimization problem using a subspace embedding approach (Wang et al., 2016; Nayebi et al., 2019; Letham et al., 2020). The main idea is to assume the existence of a low-dimensional subspace (*active subspace*) $\mathcal{Z} \subset \mathbb{R}^{d_e}$ ($d_e \leq d$), a function $g : \mathcal{Z} \to \mathbb{R}$ and a projection matrix $\boldsymbol{T} \in \mathbb{R}^{d_e \times d}$ such that $\forall \boldsymbol{x} \in \mathcal{X}, g(\boldsymbol{T}\boldsymbol{x}) = f(\boldsymbol{x})$. This property enables the optimization process to be conducted in the active subspace, which has a lower dimension compared to the original high-dimensional space. In practice, the effective dimensionality $d_e$ is generally unknown, therefore, existing approaches (Wang et al., 2016; Nayebi et al., 2019; Letham et al., 2020) randomly choose a subspace $\mathcal{V} \subset \mathbb{R}^{d_{\mathcal{V}}}$ (*target space*) to project the original space into, and perform optimization within this chosen subspace. The *target dimension* $d_{\mathcal{V}}$ must be chosen such that the probability of the target space containing the global optimum is high. In practice, choosing an appropriate value of the target dimension $d_{\mathcal{V}}$ is challenging as a small value of $d_{\mathcal{V}}$ does not guarantee that the target space contains the global optimum whilst a large value of $d_{\mathcal{V}}$ could be subject to the curse of dimensionality issue. `BAxUS` proposes an adaptive strategy to gradually increase the target dimension during the optimization process, guaranteeing a higher probability, compared to existing approaches, that its embedding contains the global optimum of the objective function. The detailed description of `BAxUS` is in Appendix Section A.2.

## 3 Related Work

Various research works have been conducted to address the challenge of applying BO to high-dimensional optimization problems. One approach is to exploit the additive structure of the objective functions, then construct and combine a large number of Gaussian Processes (GPs) to approximate the objective function (Kandasamy et al., 2015; Gardner et al., 2017). Some works suggest replacing GPs with surrogate models that might scale better with high-dimensional data such as random forests (Hutter et al., 2011), deep neural networks (Snoek et al., 2015), or Bayesian neural networks (Springenberg et al., 2016). Oh et al. (2018) propose `BOCK` whose main idea is to employ cylindrical transformation to transform the geometry of the search space, thus mitigating the over-exploration issue of BO in high-dimensional optimization problems. Other methods, such as the work by Garnett et al. (2014), `REMBO` (Wang et al., 2016), `HeSBO` (Nayebi et al., 2019), `ALEBO` (Letham et al., 2020), and `BAxUS` (Papenmeier et al., 2022) propose mapping the original high-dimensional space into a low-dimensional space and then conducting optimization within this low-dimensional space. More recently, Song et al. (2022) propose `MCTS-VS`, a meta-algorithm that employs Monte Carlo tree search (MCTS) to construct a low-dimensional subspace and then use a BO optimizer (e.g., `BO`, `TuRBO`) to optimize the objective function within this subspace.

Another approach that has recently attracted a lot of attention is the use of a local search strategy that partitions the search domain into promising local regions where the optimization can be performed within (Munos, 2011; Wang et al., 2014; Eriksson et al., 2019; Wang et al., 2020; Turner et al., 2020; Fröhlich et al., 2021; Wan et al., 2021; 2022). Notable methods in this direction include `TuRBO` (Eriksson et al., 2019), whose main idea is to construct local regions as hyper-rectangles centered around the best-found values, and dynamically expand or shrink these regions based on the function values. Wan et al. (2021) extend this idea for the categorical and mixed search space settings. Another method proposed in Wang et al. (2020), namely `LA-MCTS`, is a meta-level algorithm that uses an unsupervised K-mean algorithm to classify the search space into good and bad local regions, within which a BO optimizer such as `BO` or `TuRBO` can be employed.

Other related works, including the research by Müller et al. (2021); Nguyen et al. (2022), also employ a local search strategy. However, different compared to the local search approach of partitioning the search space into local regions, this approach aims to incorporate gradient information into the BO process to enhance its effectiveness. In particular, the work in Müller et al. (2021) develops a probabilistic model that can incorporate gradient information, and selects data points for evaluation as those that maximize the gradient. Building on this work, Nguyen et al. (2022) propose a new acquisition function designed to maximize the probability of gradient descent, thereby enabling BO to rapidly converge toward the local optimum region. The local strategies in these methods are different from the space partitioning mechanism used in `TuRBO` and `LA-MCTS`. These methods aim to perform local optimization using the current solution and seek to descend the objective function values via the gradient information computed from objective function values in nearby regions. These methods complement our proposed CMA-based meta-algorithm, as they can be used as optimizers within our approach.

Evolutionary Algorithms (EAs) represent a widely-used family of algorithms for optimizing high-dimensional black-box functions. Among these, `CMA-ES` (Hansen & Ostermeier, 2001) is well-known for its impressive performance in finding global optimum of the objective function. In this paper, we propose a novel meta-algorithm that leverages the CMA technique of `CMA-ES` to define the local regions. There are several EA methods that also combine the CMA technique with the GP surrogate model to enhance the optimization performance. `GPOP` (Buche et al., 2005) is an EA method that uses `CMA-ES` to optimize a merit function defined by the predicted mean and standard deviation function of a trained GP. `DTS-CMAES` (Bajer et al., 2019), another EA method which has been shown to outperform `GPOP`, constructs a doubly-trained GP inside `CMA-ES` to select the data points for forming the mean vector and covariance matrix of CMA. Closely related to our approach, `BADS` (Acerbi & Ma, 2017) is a global optimization method that adopts the mesh adaptive direct search framework (Audet & Dennis, 2006) which uses CMA-ES as the search oracle and a GP-based method to find the global optimum of the objective function.

In our experiments, we compare the CMA-based BO methods (created by combining our proposed CMA-based meta-algorithm with the BO optimizers `BO`, `TuRBO`, `BAxUS`) with a comprehensive list of related methods: the standard `BO` method, `TuRBO`, `BAxUS`, `LA-MCTS`, `MCTS-VS`, `CMA-ES`, `DTS-CMAES` and `BADS`.

## 4   High-dimensional Bayesian Optimization via Covariance Matrix Adaptation

In this section, we first discuss the key ideas of the CMA strategy (Section 4.1), then we propose the CMA-based meta-algorithm (Section 4.2). Finally, we derive the CMA-based BO algorithms (`CMA-BO`, `CMA-TuRBO`, `CMA-BAxUS`) corresponding to the cases when we integrate the CMA-based meta-algorithm with the BO optimizer `BO`, `TuRBO`, and `BAxUS` (Sections 4.2.1, 4.2.2, and 4.2.3).

### 4.1   The Covariance Matrix Adaptation Strategy

The CMA strategy was initially developed in the Evolutionary Algorithm literature, particularly in the `CMA-ES` method (Hansen & Ostermeier, 2001). Its main idea is based on stochastic search, which maintains a search distribution, $p(\boldsymbol{x})$, to estimate the probabilities of data points in the search domain being the global optimum of the objective function. First, a search distribution $p^{(0)}(\boldsymbol{x})$ is initialized. Then a population of data is sampled from $p^{(0)}(\boldsymbol{x})$, and the search distribution is updated based on the objective function values of these data points. This process is conducted iteratively until the algorithm converges (Hansen & Ostermeier, 2001; Abdolmaleki et al., 2017). CMA provides well-established formulas (details below) for the updates of the search distribution, enabling the stochastic search algorithm to eventually allocate the highest probability to the global optimum. It has been shown that CMA-based ES techniques, such as `CMA-ES`, perform very well in finding the global optima of high-dimensional optimization problems (Loshchilov & Hutter, 2016; Eriksson et al., 2019; Letham et al., 2020; Nomura et al., 2021).

In practice, the most popular choice for the search distribution used in CMA is the multivariate normal distribution (Hansen & Ostermeier, 2001; Hansen, 2016). Consequently, the focus of CMA is on updating the two principal moments of this distribution: the mean vector $\boldsymbol{m}$ and the covariance matrix $\boldsymbol{\Sigma}$. In CMA, the covariance matrix is normally decomposed as $\boldsymbol{\Sigma} = \sigma^2 \boldsymbol{C}$ where $\sigma > 0$ is the overall standard deviation (step size) of the search distribution, thus the goal of CMA is then to update $\boldsymbol{m}, \sigma$, and $\boldsymbol{C}$. At iteration $t$, given $\lambda$ observed data points $\{\boldsymbol{x}_i^{(t)}, y_i^{(t)}\}_{i=1}^{\lambda}$ sampled from the search distribution $\mathcal{N}\left(\boldsymbol{m}^{(t-1)}, (\sigma^{(t-1)})^2 \boldsymbol{C}^{(t-1)}\right)$ of the previous iteration $t-1$, the mean vector $\boldsymbol{m}^{(t)}$, covariance matrix $\boldsymbol{C}^{(t)}$, and step size $\sigma^{(t)}$ of the search distribution at iteration $t$ are updated as follows (Hansen & Ostermeier, 2001; Hansen, 2016),

$$\boldsymbol{m}^{(t)} = \boldsymbol{m}^{(t-1)} + c_m \sum\nolimits_{i=1}^{\mu} w_i \left(\boldsymbol{x}_{i:\lambda}^{(t)} - \boldsymbol{m}^{(t-1)}\right),$$

$$\boldsymbol{C}^{(t)} = (1 - c_1 - c_\mu) \boldsymbol{C}^{(t-1)} + \frac{c_\mu}{(\sigma^{(t-1)})^2} \sum\nolimits_{i=1}^{\lambda} w_i \left(\boldsymbol{x}_{i:\lambda}^{(t)} - \boldsymbol{m}^{(t-1)}\right) \left(\boldsymbol{x}_{i:\lambda}^{(t)} - \boldsymbol{m}^{(t-1)}\right)^{\mathsf{T}} + c_1 \boldsymbol{p}^{(t)} \boldsymbol{p}^{(t)\mathsf{T}}, \quad (4)$$

$$\sigma^{(t)} = \beta^{(t)} \sigma^{(t-1)},$$

where

- $\boldsymbol{p}^{(t)} = \sum_{i=0}^{t} (\boldsymbol{m}^{(i)} - \boldsymbol{m}^{(i-1)})/\sigma^{(i)}$ denotes the evolution path which quantifies the overall movement of the search distribution, i.e., the movement of the mean vector across iterations,

- $\boldsymbol{x}_{i:\lambda}$ denotes the $i$-th best candidate out of $\lambda$ data points $\{\boldsymbol{x}_i^{(t)}\}_{i=1}^{\lambda}$ based on their noisy function values, i.e., their corresponding noisy function values satisfy: $y_{1:\lambda}^{(t)} \leq y_{2:\lambda}^{(t)} \leq ... \leq y_{\lambda:\lambda}^{(t)}$,

- $\mu \leq \lambda$ is a hypeparameter denoting the number of data points selected to update the search distribution; by default, $\mu$ is usually set as $\lfloor \lambda/2 \rfloor$,

- $\{w_i\}_{i=1}^{\lambda}$ denotes the weight coefficients associated with the data points $\{\boldsymbol{x}_{i:\lambda}^{(t)}\}_{i=1}^{\lambda}$ such that $w_1 \geq w_2 \cdots \geq w_\mu > 0 > w_{\mu+1} \geq \cdots \geq w_\lambda$, $\sum_{i=0}^{\mu} w_i = 1$, and $\sum_{i=0}^{\lambda} w_i \approx 0$,

- $c_m$, $c_1$ and $c_\mu$ denote the learning rates at which the search distribution changes, where larger rates result in faster change and smaller rates reduce the adaptation rate of search distribution,

- $\beta^{(t)}$ denotes a modification rule for the step size, depending on the evolution path $\boldsymbol{p}^{(t)}$.

Detailed information for the suggested settings of these hyperparameters can be found in Appendix Section A.3. From Eq. (4), it can be seen that in CMA, the mean vector $\boldsymbol{m}^{(t)}$ is updated based on the previous

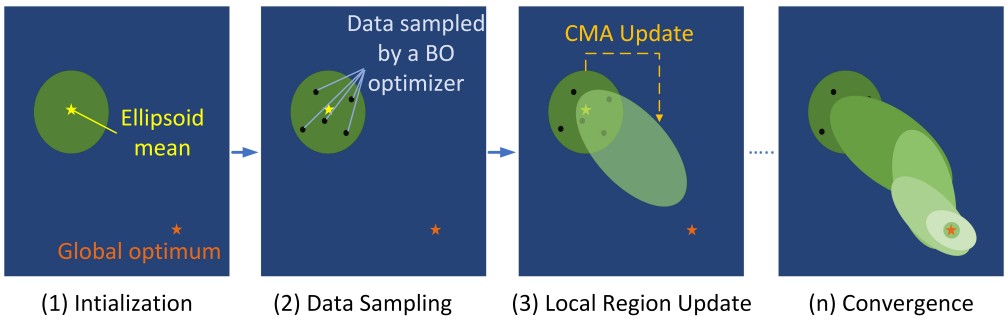

Figure 1: Illustration of the proposed CMA-based meta-algorithm. In step (1), a hyper-ellipsoid local region is initialized. In step (2), a BO optimizer (e.g., `BO`, `TuRBO`, `BAxUS`) is used in this local region to collect a population of candidates (data points to be evaluated). In step (3), the local region is updated using the CMA technique. The process is conducted iteratively until the evaluation budget is depleted.

mean vector $\boldsymbol{m}^{(t-1)}$ and the highest-ranking observed data points $\{\boldsymbol{x}_{i:\lambda}^{(t)}\}_{i=1}^{\mu}$. The covariance matrix $\boldsymbol{C}^{(t)}$ is updated based on the previous covariance matrix $\boldsymbol{C}^{(t-1)}$, all the observed data points $\{\boldsymbol{x}_{i:\lambda}^{(t)}\}_{i=1}^{\lambda}$, and the evolution path $\boldsymbol{p}^{(t)}$ of the search distribution from previous iterations. The step size $\sigma^{(t)}$ is updated based on $\beta$, which depends on the overall movement of the search distribution (the evolution path $\boldsymbol{p}^{(t)}$). If the evolution path is short, e.g., when the vectors $\Delta\boldsymbol{m}^{(i)} = \boldsymbol{m}^{(i)} - \boldsymbol{m}^{(i-1)}$ in consecutive iterations cancel each other out, the step size $\sigma$ is decreased, as the search distribution is likely to start converging toward a solution. On the contrary, when the evolution path is long, e.g., the vectors $\Delta\boldsymbol{m}^{(i)}$ are in the same direction, the step size is increased, as the search distribution is likely to be far away from the true one.

Theoretically, it has been shown that the CMA strategy can be interpreted as a natural gradient learning method that updates the parameters (mean, covariance matrix, step size) of the search distribution $p(\boldsymbol{x})$ to minimize the expected function value $\mathbb{E}[f(\boldsymbol{x})]$ under this distribution (Akimoto et al., 2010; Nomura et al., 2021). With the updates in Eq. (4), CMA tends to maximize the probability of generating successful data points $\{\boldsymbol{x}_{i:\lambda}^{(t)}\}_{i=1}^{\mu}$ (e.g., data points with lower function values for a minimization problem) in the subsequent iterations (Hansen & Auger, 2011). Empirically, as discussed at the beginning of this section, CMA-based ES techniques like `CMA-ES` perform very well in finding the global optimum of high-dimensional optimization problems (Loshchilov & Hutter, 2016; Nomura et al., 2021), demonstrating the effectiveness of CMA in deriving search distributions that can estimate the probabilities of data points being the global optimum of the objective function.

### 4.2 The CMA-based Meta-algorithm

In this section, we present our proposed CMA-based meta-algorithm. We first discuss the overall process of this meta-algorithm, then we derive three CMA-based BO methods where we integrate the proposed meta-algorithm with the three state-of-the-art BO optimizers (`BO`, `TuRBO`, `BAxUS`).

**Overall Process.** We illustrate the CMA-based meta-algorithm in Fig. 1 and the pseudocode in Algorithm 1. First, an initial search distribution $\mathcal{N}(\boldsymbol{m}^{(0)}, (\sigma^{(0)})^2 \boldsymbol{C}^{(0)})$ is set (line 6), and a local region $\mathcal{S}^{(0)}$ is computed based on this search distribution and the chosen BO optimizer `bo_opt` (line 9). Then the BO optimizer `bo_opt` is used within the local region $\mathcal{S}^{(0)}$ to suggest $\lambda$ data points $D_\lambda = \{\boldsymbol{x}_i\}_{i=1}^{\lambda}$ to be evaluated (lines 11-14). The search distribution is then updated based on these $\lambda$ observed data points $D_\lambda$ (line 15). The process is conducted iteratively until the evaluation budget is depleted, and the algorithm terminates. Note that a restart strategy is also included to restart the algorithm when the current optimization process is stuck at a local minimum (line 17).

**Local Region Formulation.** We first define the base local region $\mathcal{S}_b$ for the CMA-based meta-algorithm. Depending on the employed BO optimizer, we will further derive the local region $\mathcal{S}$ corresponding to that particular BO optimizer in the later sections (Sections 4.2.1, 4.2.2, and 4.2.3). As discussed in Section

---

**Algorithm 1** The CMA-based meta-algorithm.

---

1: **Input:** Objective function $f(.)$, search domain $[l, u]^d$, maximum number of function evaluations $N$, number of initial points $n_0$, BO optimizer `bo_opt`

2: **Output:** The optimum $\boldsymbol{x}^*$

3: Set $t \leftarrow 0$, $T \leftarrow \lfloor (N - n_0)/\lambda \rfloor$, global dataset $D \leftarrow \emptyset$, local dataset $\Omega \leftarrow \emptyset$, population size $\lambda$

4: **while** $t \leq T$ **do**

5:      Sample $n_0$ initial data points $D_0$                                                            ▷ Latin hypercube

6:      Set the initial search distribution $\mathcal{N}(\boldsymbol{m}^{(t)}, (\sigma^{(t)})^2 \boldsymbol{C}^{(t)})$ based on $D_0$                    ▷ Sec. A.5

7:      Update global dataset $D \leftarrow D \cup D_0$. Reset local dataset $\Omega \leftarrow D_0$, `restart` $\leftarrow False$

8:      **while** $t \leq T$ **and** not `restart` **do**

9:          Compute the local region $\mathcal{S}^{(t)}$ from $\mathcal{N}(\boldsymbol{m}^{(t)}, (\sigma^{(t)})^2 \boldsymbol{C}^{(t)})$ depending on `bo_opt` ▷ Eqs. (5),(6),(8)

10:          Initialize a dataset to collect $\lambda$ observed data points $D_\lambda \leftarrow \emptyset$

11:          **for** $i{=}1{:}\lambda$ **do**

12:              Apply BO optimizer `bo_opt` to propose an observed data $\{\boldsymbol{x}_i, y_i\}$ from a pool of data points

13:              Update $D_\lambda \leftarrow D_\lambda \cup \{\boldsymbol{x}_i, y_i\}$

14:          **end for**

15:          Update $\{\boldsymbol{m}^{(t+1)}, \boldsymbol{C}^{(t+1)}, \sigma^{(t+1)}\} \leftarrow \text{CMA}(\{\boldsymbol{m}^{(t)}, \boldsymbol{C}^{(t)}, \sigma^{(t)}\}, D_\lambda)$                    ▷ Eq. (4)

16:          Update $D \leftarrow D \cup D_\lambda$, $\Omega \leftarrow \Omega \cup D_\lambda$, $t \leftarrow t+1$

17:          Update `restart` $\leftarrow True$ **if** stopping criteria satisfied

18:      **end while**

19: **end while**

20: Return $\boldsymbol{x}^* = \arg\min_{\boldsymbol{x}_i \in D}\{y_i\}_{i=1}^N$ from global dataset $D = \{(\boldsymbol{x}_i, y_i)\}_{i=1}^N$

---

4.1, the search distribution by CMA can assign higher probabilities to more promising data points; therefore, we can define the local regions as the regions containing data points with high probability values from this search distribution. As CMA's search distribution is a multivariate normal distribution, we propose defining the local region as the $\alpha$-level confidence hyper-ellipsoid centered at the mean vector of this search distribution, containing $\alpha$ percent of the population of data points that follows this search distribution. Specifically, at iteration $t$, given the multivariate normal search distribution $\mathcal{N}(\boldsymbol{m}^{(t-1)}, \boldsymbol{\Sigma}^{(t-1)})$, where $\boldsymbol{\Sigma}^{(t-1)} = (\sigma^{(t-1)})^2 \boldsymbol{C}^{(t-1)}$, obtained in the previous iteration, the base hyper-ellipsoid local region $\mathcal{S}_b^{(t)}$ can be computed as,

$$\mathcal{S}_b^{(t)} = \left\{ \boldsymbol{x} \mid \Delta^{(t-1)}(\boldsymbol{x}) \leq \chi_{1-\alpha,d}^2 \right\}, \tag{5}$$

where $\Delta^{(t-1)}(\boldsymbol{x}) = \sqrt{\left(\boldsymbol{x} - \boldsymbol{m}^{(t-1)}\right)^\mathsf{T} \left(\boldsymbol{\Sigma}^{(t-1)}\right)^{-1} \left(\boldsymbol{x} - \boldsymbol{m}^{(t-1)}\right)}$ is the Mahalanobis distance (Mahalanobis, 1936) from $\boldsymbol{x}$ to the search distribution $\mathcal{N}(\boldsymbol{m}^{(t-1)}, \boldsymbol{\Sigma}^{(t-1)})$ and $\chi_{1-\alpha,d}^2$ is the Chi-squared $1 - \alpha$ critical value with $d$ degree of freedom. In our proposed CMA-based meta-algorithm, we set $\alpha$ to be 99.73%, corresponding to the 3-sigma rule that is commonly used in practice. With this setting, the selected observed data in each iteration will always fall within the three standard deviations of the mean vector of the search distribution.

**Local Optimization.** In each iteration $t$, given the multivariate normal search distribution $\mathcal{N}(\boldsymbol{m}^{(t-1)}, \boldsymbol{\Sigma}^{(t-1)})$ obtained in the previous iteration, we first sample a pool of data points that follow this search distribution and are within the local region $\mathcal{S}^{(t)}$. Then, we use the employed BO optimizer, `bo_opt`, to select $\lambda$ data points from this pool of data points. The rationale behind this step is that in the CMA strategy, the search distribution $\mathcal{N}(\boldsymbol{m}^{(t-1)}, \boldsymbol{\Sigma}^{(t-1)})$ provides estimates of the probabilities of data points in the search domain being the global optimum of the objective function. By sampling data points following this search distribution, we can have a pool of data points having high probabilities of being the global optimum. By using BO to select data points from this pool, we have a higher probability of selecting the better data points that are close to the global optimum. Finally, after obtaining $\lambda$ observed data points, we update the CMA's search distribution following Eq. (4). It is worth noting that, in our proposed approach, in each iteration, we sample and evaluate $\lambda$ data points rather than just one as in standard BO methods, i.e., in our algorithm, one iteration is equal to $\lambda$ iterations in standard BO methods.

**Restart Strategy.**  A local search strategy is typically biased toward the starting point, and the optimization process can be trapped in local minima (Eriksson et al., 2019). To enable global optimization for the CMA-based meta-algorithm, we use the CMA's restart strategy: a new local search will be initialized when the current one is stuck at a local minimum (Auger & Hansen, 2005; Hansen, 2016). The conditions for a restart in CMA normally involve checking if the objective function values are flat for a number of iterations or if some numerical indicators (e.g., condition number) of the search distribution are violated. Furthermore, since some BO optimizers (e.g., `TuRBO`, `BAxUS`) have their own restart strategies, we also incorporate these restart strategies when applying our proposed CMA-based meta-algorithm to the corresponding BO optimizers (detailed information in the subsequent sections).

### 4.2.1  CMA-BO: CMA-based Meta-algorithm with Standard BO

In this section, we describe `CMA-BO`, the corresponding CMA-based BO method obtained when incorporating the proposed CMA-based meta-algorithm with the standard `BO` optimizer. Note that from this section, we remove the superscript denoting the iteration index for brevity.

**Local Region Formulation.**  For `CMA-BO`, we define the local region $\mathcal{S}$ equal to the base local region $\mathcal{S}_b$ described in Eq. (5), i.e., the local region is the $\alpha$-level confidence hyper-ellipsoid of the search distribution $\mathcal{N}(\boldsymbol{m}, \boldsymbol{\Sigma})$. The value $\alpha$ is also set at 99.73%, corresponding to the 3-sigma rule.

**Local Optimization.**  Following the base algorithm described in Section 4.2, in each iteration, we first sample a pool of data points that (1) follow the previous search distribution $\mathcal{N}(\boldsymbol{m}, \boldsymbol{\Sigma})$, and, (2) are within the local region $\mathcal{S}$. Then we sequentially apply BO with the TS acquisition function to select the best $\lambda$ data points from this pool. Note that when training the GP, as in Eriksson et al. (2019), we also use all the observed data in all the previous iterations. The pseudocode of the local optimization step in `CMA-BO` is in Appendix Section A.4, Algorithm 2.

**Restart Strategy.**  `CMA-BO` has the same restart strategy with CMA as described in the base algorithm.

### 4.2.2  CMA-TuRBO: CMA-based Meta-algorithm with TuRBO

We derive `CMA-TuRBO`, the CMA-based BO method obtained when incorporating our CMA-based meta-algorithm with the `TuRBO` optimizer. The challenge here is that `TuRBO` has its own local region adaptation mechanism to shrink or expand. Furthermore, the shape of the local regions of `TuRBO` is hyper-rectangle which is very different from our CMA local regions' shape which is hyper-ellipsoid.

**Local Region Formulation.**  For `CMA-TuRBO`, we incorporate `TuRBO`'s local region adaptation mechanism, which is based on the success and failure state of the optimization process, with the local region strategy defined by CMA. Specifically, with the search distribution $\mathcal{N}(\boldsymbol{m}, \boldsymbol{\Sigma})$, we define the local region $\mathcal{S}_{\text{CMA-TuRBO}}$ as the hyper-ellipsoid with: (1) the center being at the mean vector $\boldsymbol{m}$, (2) the radii (lengths of the semi-axes of the hyper-ellipsoid) computed based on the covariance matrix $\boldsymbol{\Sigma}$ and scaled with a factor $L$ that is based on the success and failure state of the optimization, similar to the local region adaptation mechanism in `TuRBO`. In particular, $L$ is initially set as 0.8, and after $\tau_{\text{succ}}$ consecutive success, $L$ is doubled, while it is halved when the optimization fails to progress after $\tau_{\text{fail}}$ consecutive times. Therefore, compared to the local regions defined by `CMA-BO`, the local regions of `CMA-TuRBO` are scaled based on the historical optimization success record as in `TuRBO`. With a scale factor $L$, the local regions of `CMA-TuRBO` are defined as follows,

$$\mathcal{S}_{\text{CMA-TuRBO}} = \left\{ \boldsymbol{x} \mid \Delta_{\text{CMA-TuRBO}}(\boldsymbol{x}) \leq \chi^2_{1-\alpha,d} \right\}, \tag{6}$$

where $\Delta_{\text{CMA-TuRBO}}(\boldsymbol{x}) = \sqrt{(\boldsymbol{x}-\boldsymbol{m})^{\intercal} \boldsymbol{\Sigma}^{-1}_{\text{CMA-TuRBO}}(\boldsymbol{x}-\boldsymbol{m})}$ is the Mahalanobis distance from $\boldsymbol{x}$ to the scaled search distribution $\mathcal{N}(\boldsymbol{m}, \boldsymbol{\Sigma}_{\text{CMA-TuRBO}})$ and $\boldsymbol{\Sigma}_{\text{CMA-TuRBO}} = L^2 \boldsymbol{\Sigma}$. The derivation of this scaled covariance matrix is as follows. By definition, the radii of the hyper-ellipsoid constructed by $\boldsymbol{\Sigma}$ is $\boldsymbol{r} = \text{diag}(\boldsymbol{\Lambda}^{1/2})$, where $\boldsymbol{\Lambda}$ is the diagonal matrix whose diagonal elements are the eigenvalues of $\boldsymbol{\Sigma}$. Note that, using the eigendecomposition, we have that, $\boldsymbol{\Sigma} = \boldsymbol{U}\boldsymbol{\Lambda}\boldsymbol{U}^{-1}$ with $\boldsymbol{U}$ being the eigenvector matrix. Thus, when scaling the radii of this hyper-ellipsoid by $L$, i.e., $\boldsymbol{r}_{\text{CMA-TuRBO}} = L\boldsymbol{r}$, the covariance matrix $\boldsymbol{\Sigma}_{\text{CMA-TuRBO}}$ becomes $\boldsymbol{U}(L^2\boldsymbol{\Lambda})\boldsymbol{U}^{-1} = L^2\boldsymbol{\Sigma}$. Finally, similar to `TuRBO`, we also set upper and lower bounds for $L$, i.e., $L$ cannot exceed a threshold $L_{\max}$ and when $L$ becomes smaller than a threshold $L_{\min}$, the algorithm restarts.

**Local Optimization.** After obtaining the scaled search distribution $\mathcal{N}(\boldsymbol{m}, \boldsymbol{\Sigma}_{\texttt{CMA-TuRBO}})$ and the local region $\mathcal{S}_{\texttt{CMA-TuRBO}}$, the optimization process is conducted similarly as in the base algorithm described in Section 4.2. Specifically, we first sample a pool of data points from the search distribution $\mathcal{N}(\boldsymbol{m}, \boldsymbol{\Sigma}_{\texttt{CMA-TuRBO}})$, and then apply BO with the TS acquisition function to select $\lambda$ data points from this pool. Besides, when training the GP, as with Eriksson et al. (2019), we use all the observed data points so far. The pseudocode of the local optimization step in `CMA-TuRBO` is in Appendix Section A.4, Algorithm 3.

**Restart Strategy.** Apart from the restart strategy of CMA, we also employ the restart strategy of `TuRBO`, i.e., when $L$ shrinks below a minimum threshold $L_{\min}$, we terminate the CMA local region $\mathcal{S}_{\texttt{CMA-TuRBO}}$ and restart it at a new location randomly.

### 4.2.3 CMA-BAxUS: CMA-based Meta-algorithm with BAxUS

We present `CMA-BAxUS`, the method resulted when incorporating our proposed CMA-based meta-algorithm with the `BAxUS` optimizer. The difficulties in deriving `CMA-BAxUS` are that `BAxUS` is operated within a series of search space projections on different dimensionalities and `BAxUS` also includes the local search idea from `TuRBO` within its optimization process.

**Local Region Formulation.** As described in Section 2.4, the core idea of `BAxUS` is to perform optimization in the target space $\mathcal{V}$ of dimension $d_{\mathcal{V}}$ rather than in the original search space $\mathcal{X}$ of dimension $d$ ($d \geq d_{\mathcal{V}}$). Therefore, to incorporate `BAxUS` into our proposed CMA-based meta-algorithm, we need to compute CMA's local regions in the target space $\mathcal{V}$. Note that since `BAxUS` obtains the objective function evaluations in the original search domain $\mathcal{X}$, thus, using the CMA's update formula in Eq. (4), we can only compute the search distribution in $\mathcal{X}$. To compute the CMA's local regions in the target space $\mathcal{V}$, our main goal is to project the search distribution $\mathcal{N}_{\mathcal{X}}(\boldsymbol{m}_{\mathcal{X}}, \boldsymbol{\Sigma}_{\mathcal{X}})$ from $\mathcal{X}$ to $\mathcal{V}$, and then use the projected search distribution $\mathcal{N}_{\mathcal{V}}(\boldsymbol{m}_{\mathcal{V}}, \boldsymbol{\Sigma}_{\mathcal{V}})$ to construct the local regions. When performing function evaluation in $\mathcal{X}$, `BAxUS` uses a sparse embedding matrix $\boldsymbol{Q} : \mathcal{V} \to \mathcal{X}$, so that for any vector $\boldsymbol{v} \in \mathcal{V}$, we can compute the corresponding vector $\boldsymbol{x} \in \mathcal{X}$ as $\boldsymbol{x} = \boldsymbol{Q}\boldsymbol{v}$, and thus evaluate the objective function value $f(\boldsymbol{x})$. Our problem is then to find a projection matrix $\boldsymbol{P}$ that maps $\mathcal{X}$ to $\mathcal{V}$. This is basically a linear regression problem, and the solution can be derived as $\boldsymbol{P} = (\boldsymbol{Q}^{\intercal}\boldsymbol{Q})^{-1}\boldsymbol{Q}^{\intercal}$ (Golub & Van Loan, 2013). Having defined the linear transformation $\boldsymbol{P} : \mathcal{X} \to \mathcal{V}$, given the search distribution $\mathcal{N}_{\mathcal{X}}(\boldsymbol{m}_{\mathcal{X}}, \boldsymbol{\Sigma}_{\mathcal{X}})$, we can compute the projected search distribution $\mathcal{N}_{\mathcal{V}}(\boldsymbol{m}_{\mathcal{V}}, \boldsymbol{\Sigma}_{\mathcal{V}})$ as follows (Tong, 1990),

$$\begin{aligned} \boldsymbol{m}_{\mathcal{V}} &= \boldsymbol{P}\boldsymbol{m}_{\mathcal{X}}, \\ \boldsymbol{\Sigma}_{\mathcal{V}} &= \boldsymbol{P}\boldsymbol{\Sigma}_{\mathcal{X}}\boldsymbol{P}^{\intercal}. \end{aligned} \tag{7}$$

Furthermore, note that `BAxUS` employs `TuRBO` as their optimizer, so when defining the local regions for `BAxUS`, we also make use of the local region adaptation mechanism in `TuRBO`, which is to include a scale factor $L$ to scale the local region based on the success and failure state of the optimization process. With this, the local region $\mathcal{S}_{\mathcal{V},\texttt{CMA-BAxUS}}$ can be defined as,

$$\mathcal{S}_{\mathcal{V},\texttt{CMA-BAxUS}} = \left\{ \boldsymbol{v} \mid \Delta_{\texttt{CMA-BAxUS}}(\boldsymbol{v}) \leq \chi^2_{1-\alpha,d} \right\}, \tag{8}$$

where $\Delta_{\texttt{CMA-BAxUS}}(\boldsymbol{v}) = \sqrt{(\boldsymbol{v} - \boldsymbol{m}_{\mathcal{V}})^{\intercal} \boldsymbol{\Sigma}^{-1}_{\texttt{CMA-BAxUS}} (\boldsymbol{v} - \boldsymbol{m}_{\mathcal{V}})}$ is the Mahalanobis distance from $\boldsymbol{v} \in \mathcal{V}$ to the scaled search distribution $\mathcal{N}_{\mathcal{V}}(\boldsymbol{m}_{\mathcal{V}}, \boldsymbol{\Sigma}_{\texttt{CMA-BAxUS}})$ and $\boldsymbol{\Sigma}_{\texttt{CMA-BAxUS}} = L^2\boldsymbol{\Sigma}_{\mathcal{V}}$.

**Local Optimization.** After defining the local region $\mathcal{S}_{\mathcal{V},\texttt{CMA-BAxUS}}$ in the target space $\mathcal{V}$, we perform the optimization process similarly to the base algorithm described in Section 4.2, which is to sample a pool of data points from the search distribution $\mathcal{N}_{\mathcal{V}}(\boldsymbol{m}_{\mathcal{V}}, \boldsymbol{\Sigma}_{\texttt{CMA-BAxUS}})$ and use BO with the TS acquisition function to pick $\lambda$ data points in the target space $\mathcal{V}$. Note that, to make use of the observed data collected in previous target spaces, we employ the same splitting strategy as `BAxUS` to transform the data obtained in previous target spaces into the current target space, and add them to the observed dataset of the current target space. Finally, after collecting $\lambda$ observed data points in the target space $\mathcal{V}$, we then find the corresponding data points in the original search space $\mathcal{X}$ by using the projection $\boldsymbol{x} = \boldsymbol{Q}\boldsymbol{v}$ and evaluate their objective function values $f(\boldsymbol{x})$. From here, we can update the search distribution of CMA, $\mathcal{N}_{\mathcal{X}}(\boldsymbol{m}_{\mathcal{X}}, \boldsymbol{\Sigma}_{\mathcal{X}})$, using Eq. (4), and then recompute the local region $\mathcal{S}_{\mathcal{V},\texttt{CMA-BAxUS}}$ via the projected search distribution $\mathcal{N}_{\mathcal{V}}(\boldsymbol{m}_{\mathcal{V}}, \boldsymbol{\Sigma}_{\mathcal{V}})$. The pseudocode of the local optimization step in `CMA-BAxUS` is in Appendix Section A.4, Algorithm 4.

**Restart Strategy.** Apart from the restart strategy of CMA, we also employ the restart strategy of `BAxUS`. Specifically, when the local region of the largest target dimension shrinks smaller than the minimum threshold, i.e., when $d_\mathcal{V} = d$ and $L < L_{\min}$, we restart the embedding with target dimension $d_\mathcal{V} = d$ and identity embedding matrix $\boldsymbol{Q} = \boldsymbol{I}_d$.

## 5 Experiments

### 5.1 Experimental Setup and Baselines

We compare our proposed CMA-based meta-algorithm against a comprehensive list of related baselines, including `BO`, `TuRBO` (Eriksson et al., 2019), `BAxUS` (Papenmeier et al., 2022), `LA-MCTS` (Wang et al., 2020), `MCTS-VS` (Song et al., 2022), `CMA-ES` (Hansen & Ostermeier, 2001), `DTS-CMAES` (Bajer et al., 2019), and `BADS` (Acerbi & Ma, 2017). Since `LA-MCTS` and `MCTS-VS` are also meta-algorithms like our proposed method, we therefore compare against the corresponding methods obtained when incorporating these meta-algorithms with the `BO` and `TuRBO` optimizers, resulting in `LAMCTS-TuRBO`, `MCTSVS-BO`, `MCTSVS-TuRBO`. Note that neither `LA-MCTS` nor `MCTS-VS` provide guidance on how to incorporate `BAxUS` as a BO optimizer, so we are unable to include the `BAxUS`-based methods with these two meta-algorithms. Besides, we are also unable to include `LAMCTS-BO` as its running time is prohibitively slow on the problems used in this paper (each repeat takes approximately 3 days to run). Details of the experiment setups are in Appendix Sections A.5 and A.6. The average running time for each method are also reported in Appendix Section A.8.

### 5.2 Synthetic and Real-world Benchmark Problems

We conduct experiments on eight synthetic and three real-world benchmark problems to evaluate all methods. For synthetic problems, we use Levy-100D, Alpine-100D, Rastrigin-100D, Ellipsoid-100D, Schaffer2-100D, Branin2-500D, and two modified versions, Shifted-Levy-100D and Shifted-Alpine-100D. For real-world problems, we use Half-cheetah-102D, LassoDNA-180D and Rover-100D. These are the benchmark BO problems that used in related works including Wang et al. (2016; 2018); Eriksson et al. (2019); Nguyen et al. (2020); Wang et al. (2020); Eriksson & Poloczek (2021); Papenmeier et al. (2022); Song et al. (2022); Nguyen et al. (2022); Ziomek & Ammar (2023). Details of these problems are in Appendix Section A.7.

### 5.3 Experimental Results

#### 5.3.1 Comparison against State-of-the-art BO Optimizers

In this section, we aim to evaluate whether the proposed CMA-based meta-algorithm enhances the performance of the BO optimizers by comparing the performance of the CMA-based BO methods (`CMA-BO`, `CMA-TuRBO`, `CMA-BAxUS`) with the corresponding BO optimizers (`BO`, `TuRBO`, `BAxUS`). In Fig. 2, it can be clearly seen that our proposed CMA-based meta-algorithm significantly enhances the performance of the corresponding optimizers. `CMA-BO` and `CMA-TuRBO` outperform `BO` and `TuRBO` by a very high margin across all 11 benchmark problems. `CMA-BAxUS` outperforms `BAxUS` significantly on 6 problems (Levy-100D, Rastrigin-100D, Schaffer2-100D, Shifted-Alpine-100D, Rover-100D, LassoDNA-180D) and performs similarly on 5 problems (Alpine-100D, Ellipsoid-100D, Shifted-Levy-100D, Branin2-500D, Half-cheetah-102D). Besides, it is worth noting that, in the shifted functions, both the performance of `BAxUS` and `CMA-BAxUS` degrade drastically. This is because the global optimum is not at the center of the search domain and these methods no longer have the advantageous benefit of the sparse embedding technique. However, `CMA-BAxUS` still outperforms `BAxUS` in Shifted-Alpine-100D and has a similar performance in Shifted-Levy-100D.

#### 5.3.2 Comparison against State-of-the-art Meta-algorithms

Here, we aim to evaluate whether the proposed CMA-based meta-algorithm is better than existing state-of-the-art meta-algorithms. Specifically, we compare with other methods created by applying existing meta-algorithms to the associated BO optimizers (`LAMCTS-TuRBO`, `MCTSVS-BO`, `MCTSVS-TuRBO`). In Fig. 3, compared to existing meta-algorithms (`LA-MCTS` and `MCTS-VS`), our proposed CMA-based meta-algorithm also outperforms these state-of-the-art meta-algorithms significantly. It can be clearly seen that `CMA-TuRBO` outperforms

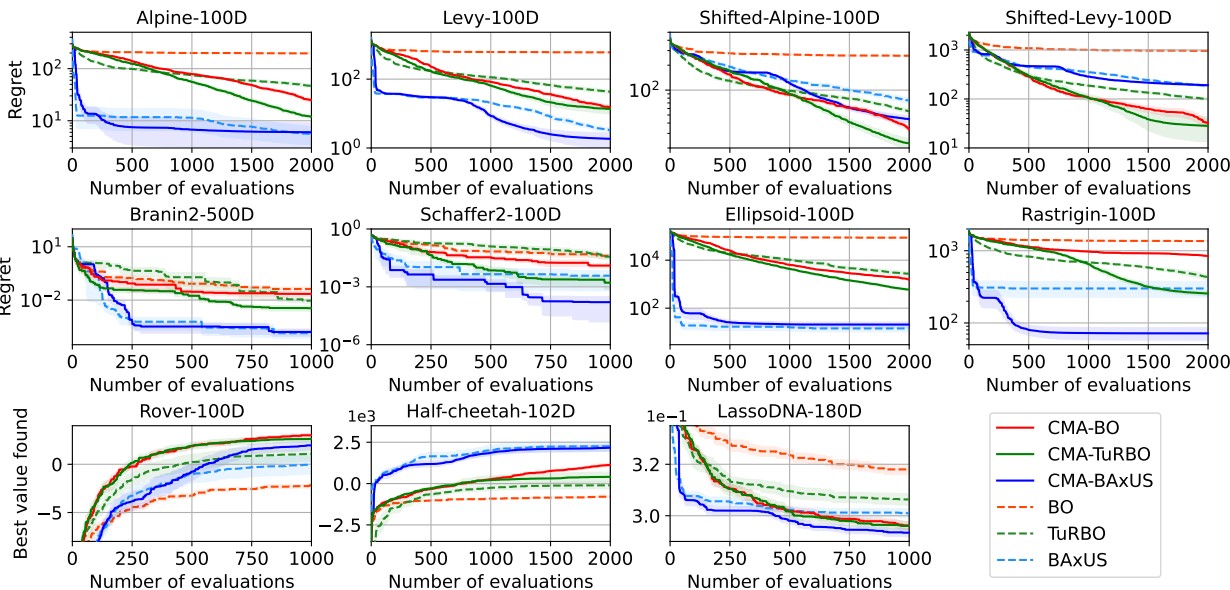

Figure 2: Comparison between the CMA-based BO methods (`CMA-BO`, `CMA-TuRBO`, `CMA-BAxUS`) against the original BO optimizers (`BO`, `TuRBO`, `BAxUS`). Plotting the mean and standard error over 10 repetitions. The CMA-based BO methods outperform their respective BO optimizers in most cases.

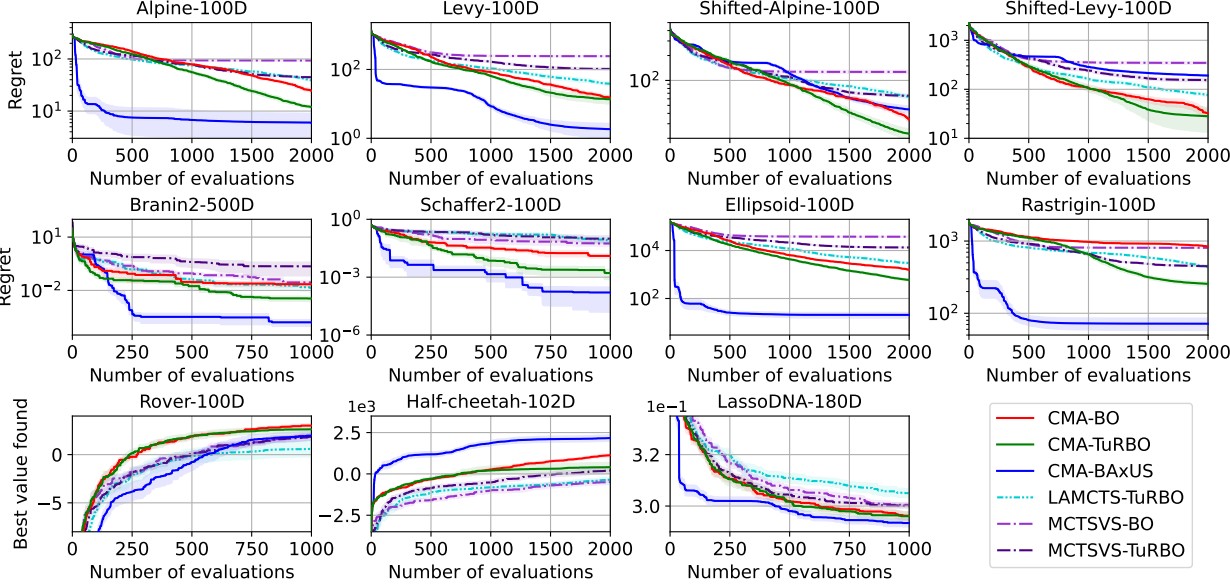

Figure 3: Comparison between the CMA-based BO methods (`CMA-BO`, `CMA-TuRBO`) against existing meta-algorithms (`LAMCTS-TuRBO`, `MCTSVS-BO`, `MCTSVS-TuRBO`). Plotting the mean and standard error over 10 repetitions. The CMA-based BO methods outperform the other meta-algorithms given the same BO optimizer.

both `LAMCTS-TuRBO` and `MCTSVS-TuRBO` by a very high margin on all of the problems. Similarly, `CMA-BO` also outperforms `MCTSVS-BO` on all of the problems by a very high margin. Note that, as mentioned in Section 5, we are unable to include `LAMCTS-BO` due to its prohibitively slow running time (approximately 3 days per one repeat). Furthermore, these meta-algorithms do not suggest on how to incorporate `BAxUS` as a BO optimizer, so we are also unable to compare them with `CMA-BAxUS`.

### 5.3.3 Comparison against other Related Baselines

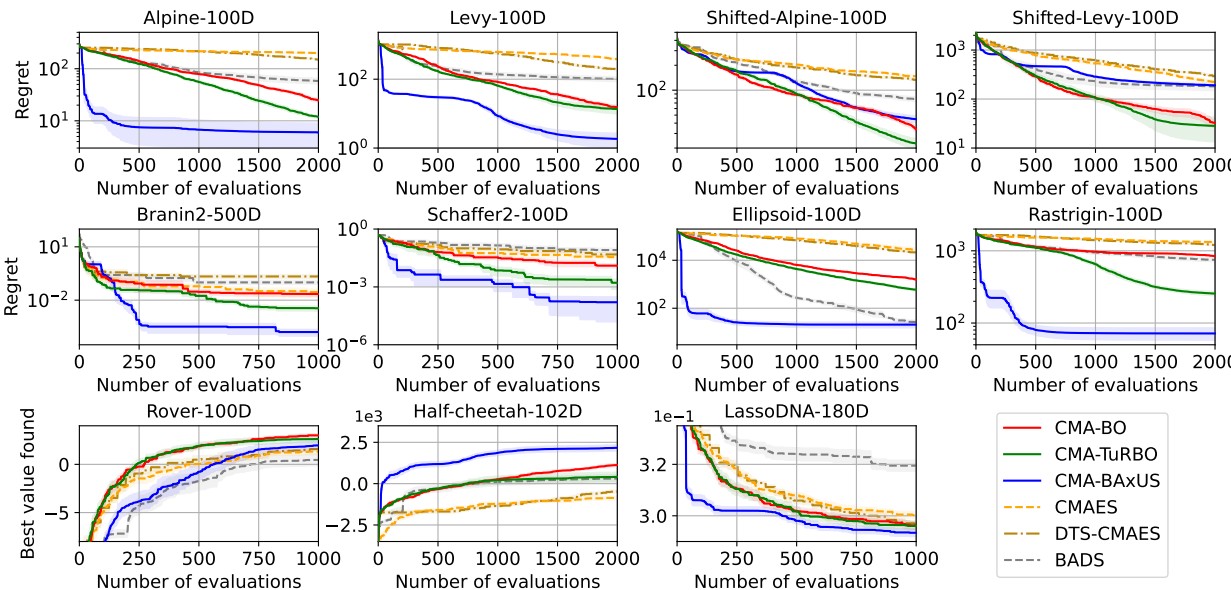

Figure 4: Comparison between the CMA-based BO methods (`CMA-BO`, `CMA-TuRBO`, `CMA-BAxUS`) against the CMA-based ES methods (`CMA-ES`, `DTS-CMAES`) and `BADS`, a global optimization method which combines `BO` and `CMA-ES`. Plotting the mean and standard error over 10 repetitions. The CMA-based BO methods outperform `CMA-ES`, `DTS-CMAES` and `BADS` consistently.

We compare the performance of our proposed CMA-based meta-algorithm with other related methods such as the CMA-based ES methods (`CMA-ES`, `DTS-CMAES`) and `BADS`, a global optimization method combining `BO` and `CMA-ES`. Fig. 4 demonstrates that the CMA-based BO methods outperform the related CMA-based ES methods in the EA literature such as `CMA-ES` and `DTS-CMAES`. This improvement could be attributed to the use of BO optimizers to select data points for the CMA strategy, instead of randomly sampling, as is the case with these evolutionary algorithms. This approach makes the CMA-based BO methods to be more data-efficient. `BADS`'s good performance on the Ellipsoid-100D problem is thanks to the directed search mechanism based on the mesh points, making `BADS` perform well on unimodal functions. However, `BADS` performance is still similar to `CMA-BAxUS` within given budget. Apart from that, `BADS` seem to struggle with the high-dimensional optimization problems with limited data and perform poorly on most of our problems.

### 5.4 Analysis of the Effectiveness of the CMA-based Meta-algorithm

### 5.4.1 The Trajectory of the Local Regions by the CMA-based Meta-algorithm

We conduct a study to understand the trajectories of the local regions defined by the CMA-based meta-algorithm in various 2D problems. Note that we only plot the local regions for two derived CMA-based BO methods, `CMA-BO` and `CMA-TuRBO`, as the behavior of `CMA-BAxUS` in 2D problems is similar to that of `CMA-TuRBO`. In Fig. 5, we show the local trajectories for the problem Shifted-Alpine-2D. Additional results for all other synthetic problems can be found in Appendix Section A.9.

We can see that at the beginning (Iteration 0), when the prior information is insufficient for BO, the selected data points scatter randomly throughout the search domain. In the later iterations, owing to the use of `BO` or `TuRBO` combining with the local regions defined by the CMA strategy, the selected data points converge closer to the global optimum. Note that in these plots, both `CMA-BO` and `CMA-TuRBO` start with the same CMA's search distribution, however, the local regions in `CMA-TuRBO` are further scaled by a factor of $L$ compared to the base local region due to its local region adaptation mechanism. In the example plotted here, the local regions of `CMA-BO` gradually shrink toward the global optimum whilst the local regions of `CMA-TuRBO` shrink

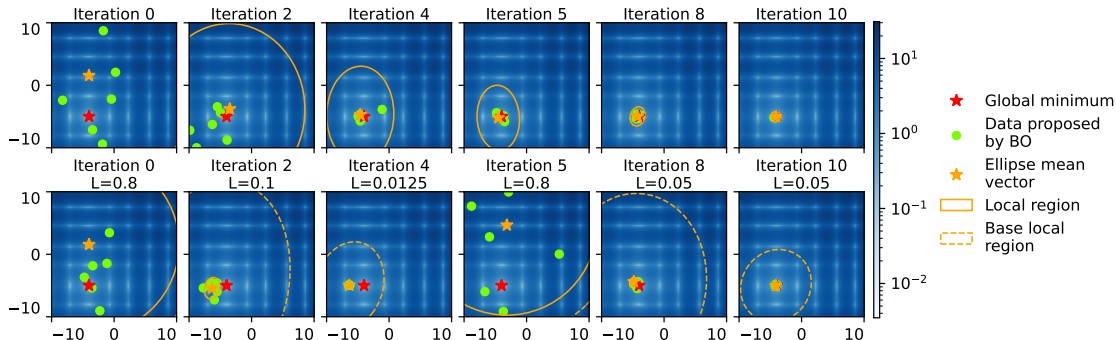

Figure 5: The local regions' trajectories defined by the proposed CMA-based meta-algorithm when paired with `BO` (upper row) and `TuRBO` (lower row) for the Shifted-Alpine-2D function. In this case, the local regions of `CMA-BO` gradually move towards the global minimum of the objective function whilst the local regions of `CMA-TuRBO` quickly converge to a sub-optimal location, then restart and move toward to the global optimum.

much faster, converge to a local optimum at Iteration 4 (with $L = 0.0125$), then restart and ultimately converge to the global optimum.

### 5.4.2 How the CMA-based Meta-algorithm Approaches the Global Optimum Compared to Baselines

In Section 5.3, we present the performance of our proposed CMA-based BO methods in terms of the best function values found. In this section, we evaluate how close the selected data points by the CMA-based BO methods are to the global optimum of the objective function compared to existing baselines. In Fig. 6, we plot the Euclidean distance between the selected data points in each iteration and the global optimum of the objective functions for all the methods. Note that we can only evaluate using the synthetic problems as it is not possible to know the global optima of the real-world problems.

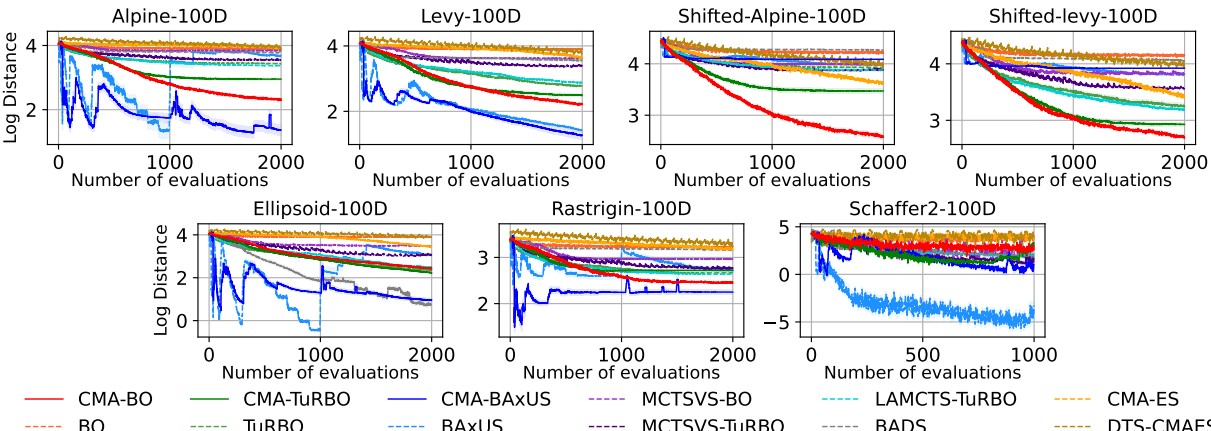

Figure 6: The Euclidean distances between selected data points in each iteration and the global optimum. The CMA-based BO methods can guide the search closer to the global optimum than other baselines.

From Fig. 6, we can see that the data points selected by the CMA-based BO methods come closer to the global optimum than other baselines. Specifically, the data points selected by `CMA-BO`, `CMA-TuRBO`, and `CMA-BAxUS` are closer to the global optimum than those selected by `BO`, `TuRBO`, and `BAxUS`, respectively in all the problems except Schaffer2-100D. Compared to other baselines, it is also clear that the CMA-based BO methods can approach closer to the global optimum in all the problems. It's worth noting that for `CMA-BAxUS`, there are some big jumps in the distance plots which is due to the changes in dimensionality of the target space, similar to `BAxUS`. When the target dimension increases, the search is performed in a higher dimension, so it needs more data to find a good solution than when in a low dimension. However,

`BAxUS` suffers this change much more than `CMA-BAxUS`, i.e., the jumps of `BAxUS` are more significant than `CMA-BAxUS`, as `CMA-BAxUS` benefits from the guidance of the CMA strategy.

### 5.4.3 How the CMA-based Meta-algorithm Locates Promising Local Regions Compared to Baselines

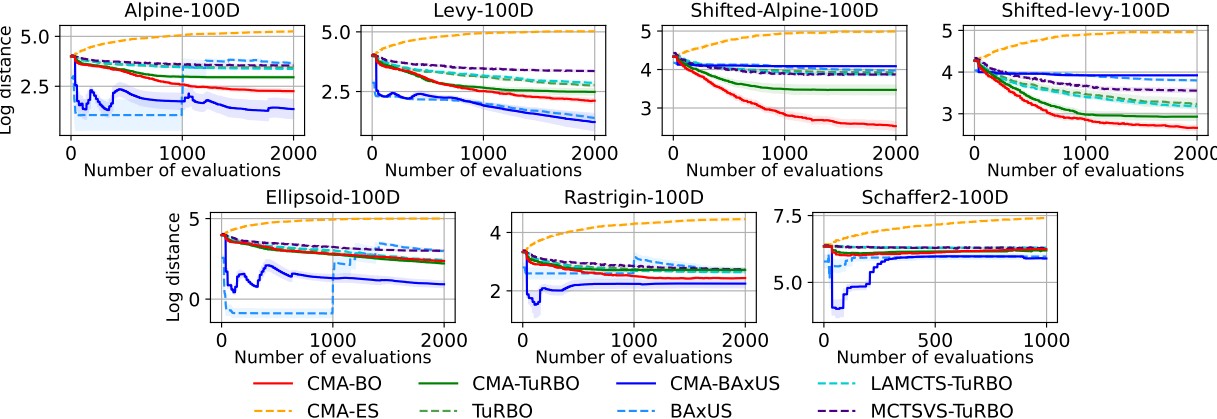

Figure 7: The Euclidean distances between centers of local regions and the global optimum. The CMA-based BO methods can guide the search closer to the global optimum compared to other baselines.

In this section, we investigate the capability of our CMA-based meta-algorithm in guiding the search closer to the promising regions that have high probabilities of containing the global optimum. We investigate the movement of the centers of the local regions defined by the methods by plotting their distances to the global optimum. We conduct this study for BO methods that define a center for their local regions, i.e., `CMA-BO`, `CMA-TuRBO`, `CMA-BAxUS`, `TuRBO`, `BAxUS`, `LAMCTS-TuRBO`, `MCTSVS-TuRBO`, and `CMA-ES`. For `CMA-BO`, `CMA-TuRBO`, `CMA-BAxUS`, and `CMA-ES`, we compute the distance between the mean vectors of the CMA search distributions at each iteration and the global optimum. For `TuRBO`, `BAxUS`, `LAMCTS-TuRBO`, and `MCTSVS-TuRBO`, we compute the distance between the centers of the hyper-rectangular local regions of these methods (defined by `TuRBO`) at each iteration and the global optimum. In Fig. 7, we can see that all the CMA-based BO methods can effectively guide their respective local regions toward the global optimum better than other baselines. `CMA-ES`, on the other hand, suffers significantly from the curse of dimensionality issue in the high-dimensional setting, and steers the search distribution away from the global optimum. This is likely due to the over-exploration of `CMA-ES`, which will be discussed in detail in Section 5.4.4. These results further confirm the capability of our proposed CMA-based meta-algorithm in identifying promising local regions.

### 5.4.4 The Effectiveness of the CMA-based BO Methods versus CMA-ES

**The Bias Issue of the CMA Updates in the CMA-based Meta-algorithm.** We investigate a key difference between the CMA-based BO methods and `CMA-ES`: the CMA update process in Eq. (4). In `CMA-ES`, the CMA search distribution is updated using $\lambda$ data points randomly sampled from the CMA multivariate normal search distribution. In our proposed CMA-based meta-algorithm, BO is used to pick $\lambda$ data points from a candidate pool of $n_c$ data points sampled from the CMA search distribution. The use of BO might introduce bias in updating the mean vector $\boldsymbol{m}$, covariance matrix $\boldsymbol{C}$, and step-size $\sigma$ in Eq. (4). One possible issue that can arise is that when $n_c$ is large, the $\lambda$ selected data points could be located near a single point. Even if $n_c$ is not so large, the CMA search distribution could be concentrated around a point, and this could cause premature convergence of the algorithm. However, we argue that in the high-dimensional setting for BO with a limited budget, this bias issue is not critical. The first reason is that the search space in a high-dimensional optimization problem is very large and thus, a standard size of $n_c$ (e.g., thousands) is not possible to make the pool of $n_c$ data points very dense, and therefore the scenario of selected data points located very close to a single point is rare when the evaluation budget is limited. The second reason is that BO has an exploitation-exploration strategy, so it does not only select data points with the best-estimated function values (exploitation), but also data points with uncertain function values

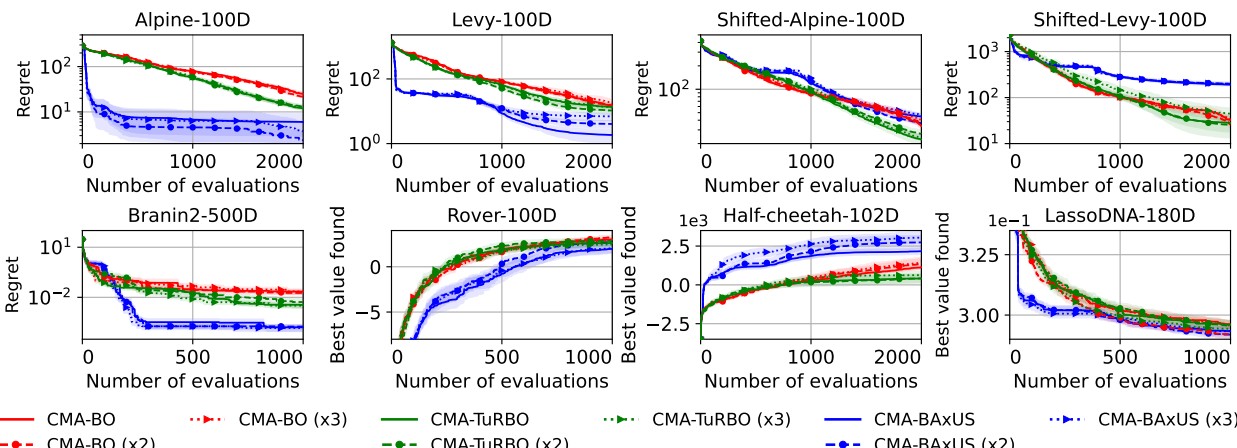

Figure 8: Performance of `CMA-BO`, `CMA-TuRBO` and `CMA-BAxUS` when increasing (double and triple) the number of sampled data points $n_c$ when optimizing acquisition function Overall, the CMA-based BO methods maintain the performance, indicating the methods' robustness w.r.t to $n_c$.

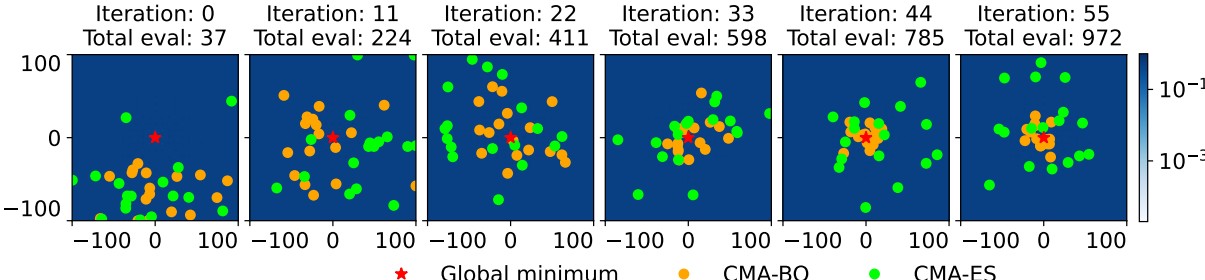

Figure 9: The 2D plot for Schaffer2-100D function projecting on the 2 effective dimensions. `CMA-ES` over-explores while `CMA-BO` can guide the search to focus on the promising region around the global optimum.

(exploration). In the high-dimensional setting, with a limited evaluation budget, the number of observed data points used to build the GP is even much smaller compared to the search space size, and this results in a GP with high uncertainty in many areas, making BO to select data points with some levels of randomness.

We conducted some analysis to validate our arguments. First, as discussed in Sections 5.4.2 and 5.4.3, the results from Figs. 6 and 7 show that the selected data points and the identified local regions by our CMA-based BO methods can approach the global optimum faster than other baselines. Furthermore, we also evaluate the robustness of our proposed CMA-based meta-algorithm w.r.t the number of sampled data points $n_c$. We increase $n_c$ to double and triple the value we use in our default setting which is $\min(100d, 5000)$, resulting in the values: $\min(200d, 10000)$ and $\min(300d, 15000)$. In Fig. 8, we plot the results of our proposed CMA-based BO methods with different values of $n_c$. We can see that the performance of our proposed methods remains similar, demonstrating their robustness to the choice of $n_c$, and thus the bias issue mentioned above is not critical to the high-dimensional setting we use in this paper.

**The Over-exploration Issue of `CMA-ES` in the High-dimensional Setting.** It is worth noting that, in practice, `CMA-ES` tends to over-explore the search space due to its random sampling strategy when selecting data points to update the search distribution. This behavior can already be seen in Fig. 6 where we can see that the data points sampled from `CMA-ES` are very far from the global optimum within our budget, and in Fig. 7 where it can be observed that the centers of the search distributions by `CMA-ES` diverge significantly from the global optimum within our evaluation budget. We further illustrate this over-exploration issue of `CMA-ES` on the Schaffer2-100D function by displaying the sampled data points of `CMA-ES` and `CMA-BO` across the optimization process. We choose Schaffer2-100D as it has 2 effective dimensions and 98 dummy

dimensions, so we can project the selected data points to these two effective dimensions and visualize the selected data points. As we have the results of 10 repeats, we present the first one in Fig. 9, and leave the remaining ones in the Appendix Section A.10. In Fig. 9, we can see that as `CMA-ES` selects data points randomly and in the high-dimensional setting, these data points are scattered everywhere in the search space, causing `CMA-ES` to over-explore. On the other hand, `CMA-BO` can select more meaningful data points, mitigating the over-exploration issue of `CMA-ES` in the high-dimensional setting.

## 6 Conclusion

In this paper, we propose a novel CMA-based meta-algorithm to address the high-dimensional BO problem by incorporating a local search strategy and the CMA strategy to enhance the performance of existing BO methods. We further derive the CMA-based BO algorithms for the cases in which our proposed meta-algorithm is applied to some common state-of-the-art BO optimizers such as `BO`, `TuRBO`, and `BAxUS`. Our extensive experimental results demonstrate the effectiveness and efficiency of the proposed CMA-based meta-algorithm, which can significantly improve the BO optimizers and outperform other state-of-the-art meta-algorithms and related methods.

### Acknowledgments

This research is supported by Australian Research Council Discovery Project DP220103044. The first and second authors (L.N. & H.H.) would like to thank the Google Cloud Research Credits Program for the computing resources on this project.

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

# A   Appendix

## A.1   TurBO

In `TuRBO`, in each iteration, the trust region (TR) is constructed as a hyper-rectangle centered at the optimum data point found so far. Each side length of the TR is initialized with a base side length $L$, and then scaled with the GP lengthscales in each dimension, while maintaining the overall hyper-volume. The size of the TR is critical, as it needs to be large enough to contain potential solutions, while being small enough to ensure the accuracy of the GP surrogate model. Therefore, `TuRBO` adopts an adaptation mechanism to expand or shrink the TR depending on whether the algorithm succeeds or fails to find a better solution. When `TuRBO` succeeds in finding better solutions for $\tau_{\text{succ}}$ consecutive times, the TR increases its current size, whereas it decreases its size after $\tau_{\text{fail}}$ consecutive failures. Furthermore, `TuRBO` also defines minimum and maximum thresholds, denoted as $L_{\min}$ and $L_{\max}$ respectively, for the TR base side length. The upper bound $L_{\max}$ is to prevent the TR from becoming too large, while the lower bound $L_{\min}$ serves as a restart criterion, such that when the side length $L < L_{\min}$, `TuRBO` discards the current TR and restarts a new TR from scratch. These hyperparameters (e.g., $\tau_{\text{succ}}$, $\tau_{\text{fail}}$, $L_{\min}$, $L_{\max}$) are set as some fixed values in `TuRBO`.

### A.2 BAxUS

BAxUS starts with a low value of the target dimension $d_y$, and then conducts a BO process to search for the optimum in this target space within a specific evaluation budget before increasing the target dimension. Additionally, BAxUS also employs TuRBO as its BO optimizer to perform optimization for high-dimensional problems. Therefore, in each iteration, BAxUS also constructs a TR and applies the TR adaptation mechanism, similar to TuRBO, when optimizing within the target space. BAxUS keeps most of the settings to be the same with TuRBO (e.g., hyper-rectangle TR, shrinkage factor, success tolerance), but redefines the failure tolerance $\tau_{\text{fail}}$ to make the search quicker in low-dimensional target spaces.

### A.3 Additional Information for the CMA Update Formula

Here, we provide additional information about how to set the hyperparmeters for the CMA formula in Eq. (4) based on Hansen & Ostermeier (2001). Given the problem dimension as $d$ and the population size $\lambda = 4 + \lfloor 3 + \ln d \rfloor$, let us define some additional terms as,

$$
\begin{aligned}
w_i' &= \ln \frac{\lambda + 1}{2} - \ln i, \text{ for } i = 1, \ldots, \lambda, \\
\mu_{\text{eff}} &= \frac{(\sum_{i=1}^{\mu} w_i')^2}{\sum_{i=1}^{\mu} w_i'^2}, \\
\mu_{\text{eff}}^- &= \frac{(\sum_{i=\mu+1}^{\lambda} w_i')^2}{\sum_{i=\mu+1}^{\lambda} w_i'^2}.
\end{aligned}
\tag{9}
$$

The learning rates coefficient are as follows,

$$
\begin{aligned}
c_m &= 1, \\
c_1 &= \frac{2}{(d + 1.3)^2 + \mu_{\text{eff}}}, \\
c_\mu &= \min \left( 1 - c_1, 2 \frac{\mu_{\text{eff}} - 2 + 1/\mu_{\text{eff}}}{(d + 2)^2 + \mu_{\text{eff}}} \right).
\end{aligned}
\tag{10}
$$

The weight coefficients $w_i$ is set as,

$$
w_i = \begin{cases}
\frac{1}{\sum_{i=1}^{\mu} w_i'} w_i' & \text{if } w_i' > 0, \\
\min \left( 1 + \frac{c_1}{c_\mu}, 1 + \frac{2\mu_{\text{eff}}^-}{2 + \mu_{\text{eff}}}, \frac{1 - c_1 - c_\mu}{nc_\mu} \right) \frac{1}{\sum_{i=\mu+1}^{\lambda} w_i'} w_i' & \text{if } w_i' < 0.
\end{cases}
\tag{11}
$$

Regarding the the covariance matrix update (second line in Eq. (4)), in practice, the evolution path $\boldsymbol{p}^{(t)} = \sum_{i=0}^{t} (\boldsymbol{m}^{(i)} - \boldsymbol{m}^{(i-1)})/\sigma^{(i)}$ is computed via exponential smoothing. Initialized with $\boldsymbol{p}^{(0)} = 0$, the exact formula of the evolution path is as follows,

$$
\boldsymbol{p}^{(t)} = (1 - c_c)\boldsymbol{p}^{(t-1)} + \sqrt{c_c(2 - c_c)\mu_{\text{eff}}} \frac{\boldsymbol{m}^{(t)} - \boldsymbol{m}^{(t-1)}}{\sigma^{(t-1)}},
\tag{12}
$$

where

$$
c_c = \frac{4 + \mu_{\text{eff}}/d}{d + 4 + 2\mu_{\text{eff}}/d}.
\tag{13}
$$

Regarding the $\beta$ coefficient in the step size update (third line in Eq. (4)), the exact formula is as follows,

$$
\beta^{(t)} = \exp \left( \frac{c_\sigma}{d_\sigma} \left( \frac{\|\boldsymbol{p}_\sigma^{(t)}\|}{\sqrt{d}} - 1 \right) \right),
\tag{14}
$$

where

$$
\begin{aligned}
c_\sigma &= \frac{2 + \mu_{\text{eff}}}{d + 5 + 2\mu_{\text{eff}}}, \\
d_\sigma &= 1 + 2\max\left(0, \sqrt{\frac{\mu_{\text{eff}-1}}{d+1}} - 1\right) + c_\sigma, \\
\boldsymbol{p}_\sigma^{(t)} &= (1 - c_c)\boldsymbol{p}_\sigma^{(t-1)} + \sqrt{c_c(2 - c_c)\mu_{\text{eff}}}\boldsymbol{C}^{(t-1)^{-\frac{1}{2}}}\frac{\boldsymbol{m}^{(t)} - \boldsymbol{m}^{(t-1)}}{\sigma^{(t-1)}}, \text{ with } \boldsymbol{p}_\sigma^{(0)} = 0.
\end{aligned}
\tag{15}
$$

### A.4 Pseudocode of the CMA-based BO Algorithms

We present the pseudocode for the local optimization steps of `CMA-BO`, `CMA-TuRBO` and `CMA-BAxUS`. These are the detailed implementation of line 12 in Algorithm 1 depending on the BO optimizer `bo_opt`. Note that, as discussed in the base algorithm in Section 4.2, when performing the local optimization step, we first need to sample a pool of $n_c$ data points following the previous search distribution, then perform BO to select the data points. In practice, we set $n_c = \min(100d_c, 5000)$ where $d_c = d$, the dimensionality of the problem, for `CMA-BO` and `CMA-TuRBO` or $d_c = d_\mathcal{V}$, the current target dimensionality, for `CMA-BAxUS`. This value of $n_c$ is the same as in `TuRBO` when selecting sampling data points for the TS acquisition function.

---

**Algorithm 2** Local Optimization for `CMA-BO`.

---

1: **Input:** Objective function $f(.)$, search distribution $\mathcal{N}(\boldsymbol{m}, \boldsymbol{\Sigma})$, local region $\mathcal{S}$, dataset $\Omega$, number of sampling points $n_c$
2: **Output:** A new observed data $\{\boldsymbol{x}_{\text{next}}, y_{\text{next}}\}$
3: Train a GP from $\Omega$
4: Sample $n_c$ data points $\mathcal{A} = \{\boldsymbol{x}_j\}_{j=1}^{n_c}$ from $\mathcal{N}(\boldsymbol{m}, \boldsymbol{\Sigma})$ and constrained within $\mathcal{S}$
5: Propose a next observed data $\boldsymbol{x}_{\text{next}} = \arg\min_{\boldsymbol{x} \in \mathcal{A}} \alpha^{\text{TS}}(\boldsymbol{x})$
6: Evaluate the observed data $y_{\text{next}} = f(\boldsymbol{x}_{\text{next}}) + \varepsilon$
7: Return $\{\boldsymbol{x}_{\text{next}}, y_{\text{next}}\}$

---

---

**Algorithm 3** Local Optimization for `CMA-TuRBO`.

---

1: **Input:** Objective function $f(.)$, search distribution $\mathcal{N}(\boldsymbol{m}, \boldsymbol{\Sigma}_{\text{CMA-TuRBO}})$, local region $\mathcal{S}_{\text{CMA-TuRBO}}$, dataset $\Omega$, number of sampling points $n_c$
2: **Output:** A new observed data $\{\boldsymbol{x}_{\text{next}}, y_{\text{next}}\}$
3: Train a GP from $\Omega$
4: Sample $n_c$ data points $\mathcal{A} = \{\boldsymbol{x}_j\}_{j=1}^{n_c}$ from $\mathcal{N}(\boldsymbol{m}, \boldsymbol{\Sigma}_{\text{CMA-TuRBO}})$ and constrained within $\mathcal{S}_{\text{CMA-TuRBO}}$
5: Propose a next observed data $\boldsymbol{x}_{\text{next}} = \arg\min_{\boldsymbol{x} \in \mathcal{A}} \alpha^{\text{TS}}(\boldsymbol{x})$
6: Evaluate the observed data $y_{\text{next}} = f(\boldsymbol{x}_{\text{next}}) + \varepsilon$
7: Return $\{\boldsymbol{x}_{\text{next}}, y_{\text{next}}\}$

---

---

**Algorithm 4** Local Optimization for `CMA-BAxUS`.

---

1: **Input:** Objective function $f(.)$, search distribution $\mathcal{N}_\mathcal{V}(\boldsymbol{m}_\mathcal{V}, \boldsymbol{\Sigma}_{\text{CMA-BAxUS}})$, local region $\mathcal{S}_{\mathcal{V},\text{CMA-BAxUS}}$, dataset $\Omega$, number of sampling points $n_c$, embedding matrix $\boldsymbol{Q} : \mathcal{V} \to \mathcal{X}$
2: **Output:** A new observed data $\{\boldsymbol{x}_{\text{next}}, \boldsymbol{v}_{\text{next}}, y_{\text{next}}\}$ in both $\mathcal{X}$ and $\mathcal{V}$
3: Train a GP from $\{\boldsymbol{v}_i, y_i\}_{i=1}^{|\Omega|} \in \Omega$
4: Sample $n_c$ data points $\mathcal{A} = \{\boldsymbol{v}_j\}_{j=1}^{n_c}$ from $\mathcal{N}(\boldsymbol{m}_\mathcal{V}, \boldsymbol{\Sigma}_{\text{CMA-BAxUS}})$ and constrained within $\mathcal{S}_{\mathcal{V},\text{CMA-BAxUS}}$
5: Propose a next observed data $\boldsymbol{v}_{\text{next}} = \arg\min_{\boldsymbol{v} \in \mathcal{A}} \alpha^{\text{TS}}(\boldsymbol{v})$ and $\boldsymbol{x}_{\text{next}} = \boldsymbol{Q}\boldsymbol{v}_{\text{next}}$
6: Evaluate the observed data $y_{\text{next}} = f(\boldsymbol{x}_{\text{next}}) + \varepsilon$
7: Return $\{\boldsymbol{x}_{\text{next}}, \boldsymbol{v}_{\text{next}}, y_{\text{next}}\}$

---

## A.5 Experimental Setup

We use Matérn 5/2 ARD kernels for the GPs in all methods. The input domains of all problems are scaled to have equal domain lengths in all dimensions as in Loshchilov & Hutter (2016). The output observations are normalized following a Normal distribution $y \sim \mathcal{N}(0,1)$.

For the hyperparameters of the CMA strategy in all CMA-based BO and ES methods, we set them using the suggested values in Hansen (2016). Specifically, the population size $\lambda$ is set to be $4 + \lfloor 3 + \ln d \rfloor$. The initial mean vector $\boldsymbol{m}^{(0)}$ is selected by minimizing 20 initial data points following a Latin hypercube sampling (Jones, 2001). The covariance matrix $\boldsymbol{C}^{(0)}$ is initialized with an identity matrix $\boldsymbol{I}_d$, and the initial step size $\sigma^{(0)}$ is set to $0.3(u-l)$ where $u, l$ denote the upper and lower bounds of the search domain $\mathcal{X}$, i.e., $\mathcal{X} = [l, u]^d$.

To ensure fair comparison between the CMA-based BO methods and the corresponding BO optimizers, we set the hyperparameters of the CMA-based BO methods to be the same as those of the corresponding BO optimizers. Specifically, for `BO` and `CMA-BO`, the hyperparameter settings of the GP and the TS acquisition function of these methods are the same. For `TuRBO` and `CMA-TuRBO`, we follow the same setting suggested by `TuRBO` (Eriksson et al., 2019) to set the initial TR base side length $L_0$, the maximum and minimum TR side lengths $L_{\max}$ and $L_{\min}$, and the success and failure threshold $\tau_{\text{succ}}$ and $\tau_{\text{fail}}$. For `BAxUS` and `CMA-BAxUS`, we also follow the same setting suggested by `BAxUS` (Papenmeier et al., 2022) to set the initial TR base side length $L_0$, the maximum and minimum TR side lengths $L_{\max}$ and $L_{\min}$, the success and failure thresholds $\tau_{\text{succ}}$ and $\tau_{\text{fail}}$, and the bin size $b$. All the developed CMA-based BO methods (`CMA-BO`, `CMA-TuRBO`, `CMA-BAxUS`) are implemented using GPyTorch (Gardner et al., 2018) as with `TuRBO` and `BAxUS`. All the Python-based methods are run with the same Python package versions.

## A.6 Baselines

To evaluate the baseline methods described in Section 5, we use the implementation and hyperparameter settings provided in the authors' public source code and their respective papers. Note that for `DTS-CMAES` and `BADS`, the authors' implementation source code is in Matlab, so to ensure consistency in the objective function evaluation process with other baselines, we call Python from Matlab to evaluate the objective function values. All the methods are initialized with 20 initial data points and are run for 10 repeats with different random seeds. All experimental results are averaged over these 10 independent runs. We then report the mean and the standard error of the simple regret or the best optimal value found. Details of the implementation for each baseline in the paper are as follows.

**BO.** This is the standard `BO` method with the TS acquisition function. The GP is constructed with the Matérn 5/2 ARD kernel and is fitted using the Maximum Likelihood method. The domains of the input variables in all problems are scaled to have equal domain lengths in all dimensions (Loshchilov & Hutter, 2016). The output observations $\{y_i\}$ are normalized following a Normal distribution $\mathcal{N}(0,1)$.

**TuRBO (Eriksson et al., 2019).** We set all the hyperparameters of `TuRBO` as suggested in their paper. This includes the upper and lower bound for TR side length $L_{\max} = 1.6$, $L_{\min} = 2^{-7}$, batch size $b = 1$ and the TR adaptation threshold $\tau_{\text{succ}} = 3$, $\tau_{\text{fail}} = \lceil \max(4/b, d/b) \rceil$ where $d$ is the dimension of the problem. We use their implementation that is made available at `https://github.com/uber-research/TuRBO`.

**BAxUS (Papenmeier et al., 2022).** We set all the hyperparameters of `BAxUS` as suggested in their paper. This includes the upper and lower bound for TR side length $L_{\max} = 1.6$, $L_{\min} = 2^{-7}$, the TR adaptation threshold $\tau_{\text{succ}} = 3$ and bin size $b = 3$, budget to input dim $m_D$ is set to the maximum budget. We use their implementation that is made available at `https://github.com/LeoIV/BAxUS`.

**LA-MCTS (Wang et al., 2020).** We set all the hyperparameters of `LA-MCTS` as suggested in their paper. This includes the exploration factor in UCB $C_p = 1$, the kernel type of SVM is RBF and the splitting threshold $\theta = 20$. For Levy function, we use different settings, which is recommended in the author's

implementation code[1], i.e., $C_p = 10$, polynomial kernel, $\theta = 8$. We use their implementation that is made available at `https://github.com/facebookresearch/LaMCTS`.

**MCTSVS (Song et al., 2022).** We set all the hyperparameters of `MCTS-VS` as suggested in their paper. This includes the exploration factor in UCB $C_p = 1$, the fill-in strategy of "best-k" with $k = 20$, the feature batch size $N_v = 2$, the sample batch size $N_s = 3$, the tree re-initialization threshold $N_{bad} = 5$, the node splitting threshold $N_{split} = 3$. We use their implementation that is made available at `https://github.com/lamda-bbo/MCTS-VS`.

**CMAES (Hansen & Ostermeier, 2001).** We use the default settings as suggested in the paper, which is similar to our settings for `CMA-BO`. This includes the population size $\lambda = 4 + \lfloor 3 + \ln d \rfloor$ where $d$ is the problem dimension, the random initial mean vector $\boldsymbol{m}^{(0)}$ selected from the minimum of the 20 random initial points, the identity initial covariance matrix $\boldsymbol{C}^{(0)} = \boldsymbol{I}_d$ and the initial step-size $\sigma^{(0)} = 0.3(u - lb)$ where the domain is scaled to uniform bound of $[l, u]^d$. We also activate the restart mechanism of `CMA-ES` so that the algorithm can restart when it converges to a local minimum. We use their implementation that is made available at `https://github.com/CMA-ES/pycma`.

**DTS-CMAES (Bajer et al., 2019).** We set all the hyperparameters of `DTS-CMAES` as suggested in their paper. We use the doubly-trained GP configuration with the population size $\lambda = 8 + \lfloor 6 + \ln d \rfloor$ where $d$ is the problem dimension, initial mean vector $\boldsymbol{m}^{(0)}$ selected from the minimum of the 20 random initial points, the identity initial covariance matrix $\boldsymbol{C}^{(0)} = \boldsymbol{I}_d$ and the initial step-size $\sigma^{(0)} = 0.3(u - lb)$ where the domain is scaled to uniform bound of $[l, u]^d$ and fixed learning rate $\beta = 0.05$. We use their implementation that is made available at `https://github.com/bajeluk/surrogate-cmaes`.

**BADS (Acerbi & Ma, 2017)** We set all the hyperparameters of `BADS` as suggested in their paper and the Matlab package. We use their implementation that is made available at `https://github.com/acerbilab/bads`.

### A.7 Synthetic and Real-world Benchmark Problems

We conduct experiments on eight synthetic and three real-world benchmark problems to evaluate all methods.

**Synthetic Problems.** We use Levy-100D, Alpine-100D, Rastrigin-100D, Ellipsoid-100D, Schaffer2-100D, Branin2-500D, and two modified versions, Shifted-Levy-100D and Shifted-Alpine-100D. For Branin2-500D, we use the implementation from Papenmeier et al. (2022); Wang et al. (2016) where the function is created by adding additional 498 dummy dimensions to the original Branin 2D function, resulting in a function with the dimension $d$ to be 500. Schaffer2-100D is implemented similarly with 98 dummy dimensions added to the original Schaffer 2D function. The Levy-100D, Alpine-100D, Rastrigin-100D and Ellipsoid-100D are common test functions[2] used in BO research, and we set the dimension $d$ to be 100 for each function. Additionally, for Levy-100D and Alpine-100D, we create two new versions, namely Shifted-Alpine-100D and Shifted-Levy-100D, where we shift the global optimum away from the original global optimum by uniformly random shifting, i.e., we set $f_{\text{shifted}}(\boldsymbol{x}) = f_{\text{original}}(\boldsymbol{x} + \boldsymbol{\delta})$ with $\boldsymbol{\delta} = [\delta_1, \ldots, \delta_d] \in [l, u]^{100}$ and $\delta_i \sim \mathcal{U}(l, u)$. The search domains of these two functions, $\mathcal{X} = [l, u]^d$, are kept the same as in the original functions. The motivation behind including these two shifted synthetic problems for evaluation is that, based on our observations, some sparse embedding methods (e.g., `BAxUS`) have considerable advantages when the global optimum is at the center of the search domain, thus, we also evaluate all methods on problems where the global optima are not at the search domain's center.

**Real-world Problems.** We use the following real-world problems: Half-cheetah-102D, LassoDNA-180D and Rover-100D. For Half-cheetah-102D, we use the same implementation as described in Song et al. (2022),

---

[1] `https://github.com/facebookresearch/LaMCTS/blob/489bd60886f23b0b76b10aa8602ea6722f334ad6/LA-MCTS/functions/functions.py`

[2] `https://www.sfu.ca/~ssurjano/optimization.html`

which parameterizes the Half-cheetah-v4 Mujoco environment from the Gym package[3] into a 102D reinforcement learning (RL) problem. The goal of this problem is to optimize the parameters of a linear policy designed to solve the RL task. These Mujoco RL tasks have been used in many works, such as Wang et al. (2020); Nguyen et al. (2020); Song et al. (2022). For LassoDNA-180D, we use the implementation from the Python LassoBench library (Šehić et al., 2022) as in Papenmeier et al. (2022). The problem LassoDNA-180D solves the Least Absolute Shrinkage and Selection Operator (LASSO) problem using the DNA dataset from a microbiology problem. The LassoBench suite has been used in several previous works, such as Papenmeier et al. (2022); Ziomek & Ammar (2023). For Rover-100D, we use the implementation provided by Wang et al. (2018). This problem optimizes the locations of 50 points in a 2D-plane trajectory of a rover, resulting in a 100D benchmark function. The goal is to maximize the reward calculated based on the number of collisions along the rover trajectory. The Rover function has been used in various research works, e.g., Wang et al. (2018); Eriksson et al. (2019); Eriksson & Poloczek (2021); Nguyen et al. (2022).

### A.8 Running time of all methods

We report the average running time per each iteration in Table 1. The results demonstrate that the running time of our proposed CMA-based BO methods is very similar to the running time of the BO optimizers we incorporate. This demonstrate the efficacy of our proposed meta-algorithm.

Table 1: Average time (in second) for each iteration run in each method.

| Average time per iteration (s) | Alpine 100D | Levy 100D | Shifted Alpine 100D | Shifted Levy 100D | Ellipsoid 100D | Rastrigin 100D | Schaffer2 100D | Branin2 500D |
|---|---|---|---|---|---|---|---|---|
| CMA-BO | 3.3 | 3.28 | 3.62 | 3.46 | 3.21 | 5.48 | 3.07 | 6.8 |
| CMA-TuRBO | 3.13 | 3.17 | 3.33 | 3.34 | 3.24 | 3.17 | 2.98 | 9.19 |
| CMA-BAxUS | 8.68 | 21.27 | 21.28 | 21.4 | 8.27 | 9.32 | 5.93 | 15.19 |
| BO | 5.7 | 5.69 | 5.63 | 5.61 | 5.57 | 5.6 | 2.57 | 1.73 |
| TuRBO | 5.32 | 5.29 | 5.28 | 5.53 | 5.39 | 5.45 | 2.46 | 2.97 |
| BAxUS | 9.61 | 21.79 | 21.42 | 20.62 | 7.56 | 8.61 | 7.04 | 9.36 |
| MCTSVS-BO | 0.96 | 0.36 | 1.5 | 1.18 | 0.48 | 0.63 | 0.25 | 0.43 |
| MCTSVS-TuRBO | 0.24 | 0.27 | 0.79 | 0.8 | 0.27 | 0.23 | 0.13 | 0.21 |
| LAMCTS-TuRBO | 3.53 | 3.51 | 10.03 | 8.99 | 3.46 | 3.45 | 1.95 | 2.39 |
| CMA-ES | 4.6E-04 | 3.2E-03 | 4.4E-04 | 2.9E-03 | 4.4E-04 | 1.7E-03 | 4.7E-04 | 1.0E-03 |
| BADS | 1.52 | 1.74 | 1.78 | 1.6 | 1.18 | 1.7 | 0.28 | 0.29 |
| DTS-CMAES | 0.06 | 0.05 | 0.05 | 0.05 | 0.06 | 0.05 | 0.03 | 0.44 |

### A.9 Additional Trajectory Plots of the Local Regions by the CMA-based Meta-algorithm

We show the remaining trajectory plots of the local regions defined by CMA-BO and CMA-TuRBO in the remaining synthetic functions: Alpine-2D (Fig. 10), Levy-2D (Fig. 11), Shifted-Levy-2D (Fig. 12), Branin-2D (Fig. 13), Ellipsoid-2D (Fig. 14), Schaffer-2D (Fig. 15) and Rastrigin-2D (Fig. 16).

### A.10 Additional Results of Schaffer2-100D

As an expansion of Fig. 9, we show in Fig. 17 all the repeats of the 2D plot for Schaffer2-100D functions.

---

[3]https://www.gymlibrary.dev/index.html

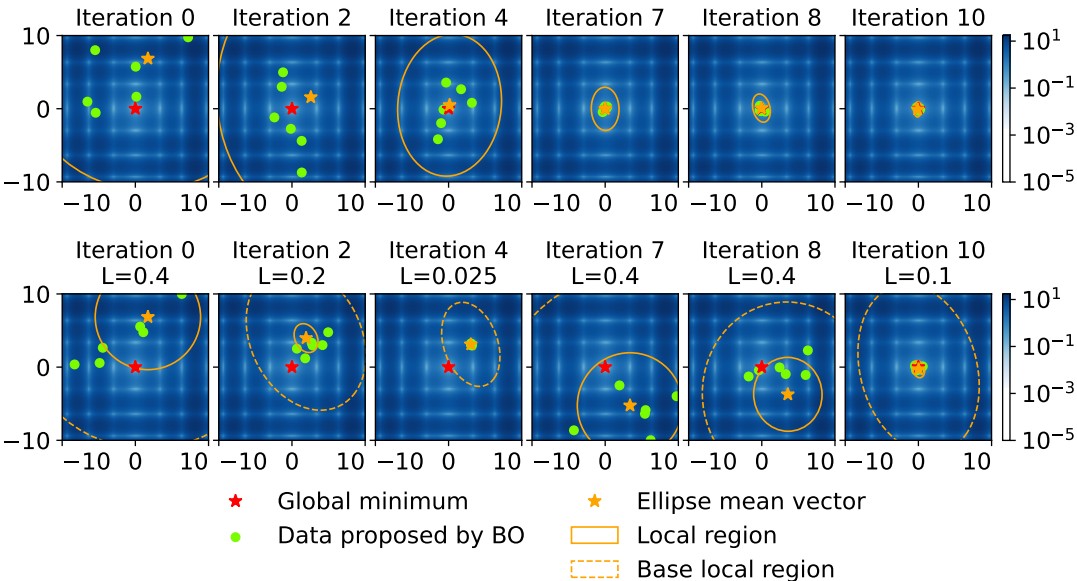

Figure 10: Trajectories of the local regions defined by CMA-based BO methods, `CMA-BO` (upper) and `CMA-TuRBO` (lower), for Alpine 2D function. The function global optimum is at $[0, 0]$.

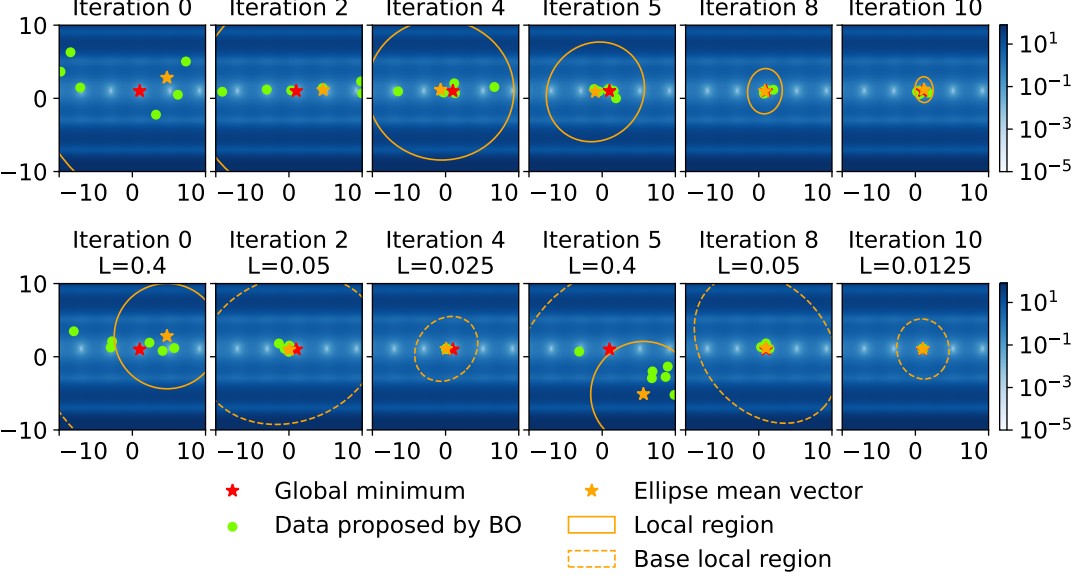

Figure 11: Trajectories of the local regions defined by CMA-based BO methods, `CMA-BO` (upper) and `CMA-TuRBO` (lower), for Levy 2D function. The function global optimum is at $[1, 1]$.

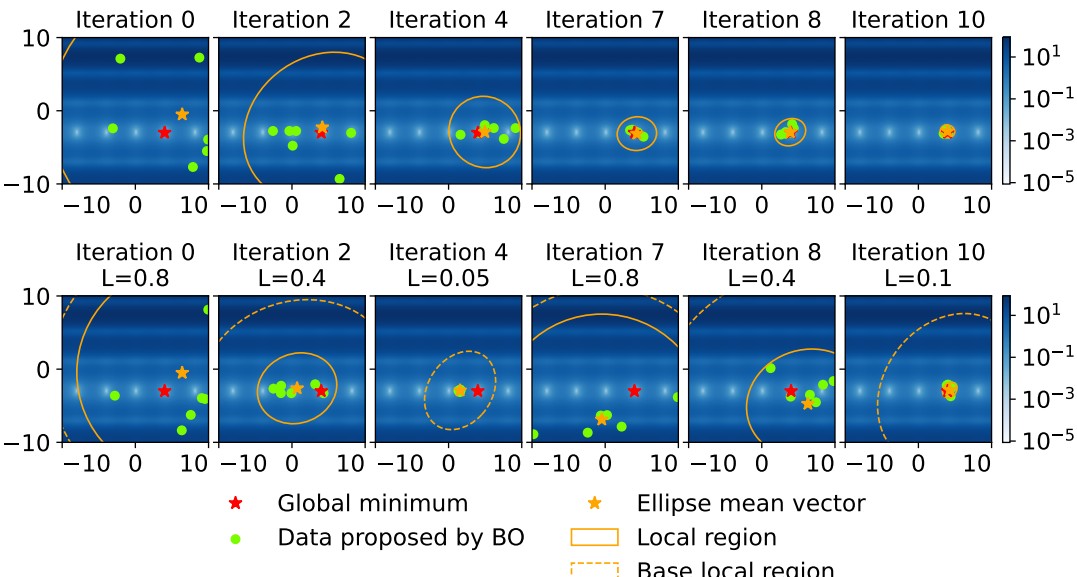

Figure 12: Trajectories of the local regions defined by CMA-based BO methods, `CMA-BO` (upper) and `CMA-TuRBO` (lower), for Shifted-Levy 2D function. The function global optimum is at $[3, -4]$.

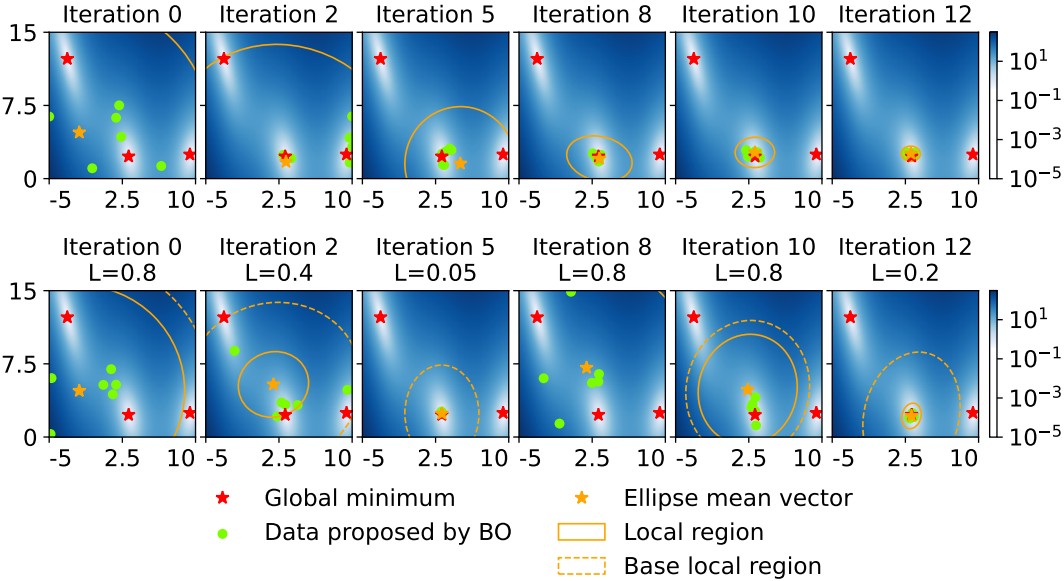

Figure 13: Trajectories of the local regions defined by CMA-based BO methods, `CMA-BO` (upper) and `CMA-TuRBO` (lower), for Branin 2D function. The function has 3 global optima at $[-\pi, 12.275]$, $[\pi, 2.275]$ and $[9.42478, 2.475]$.

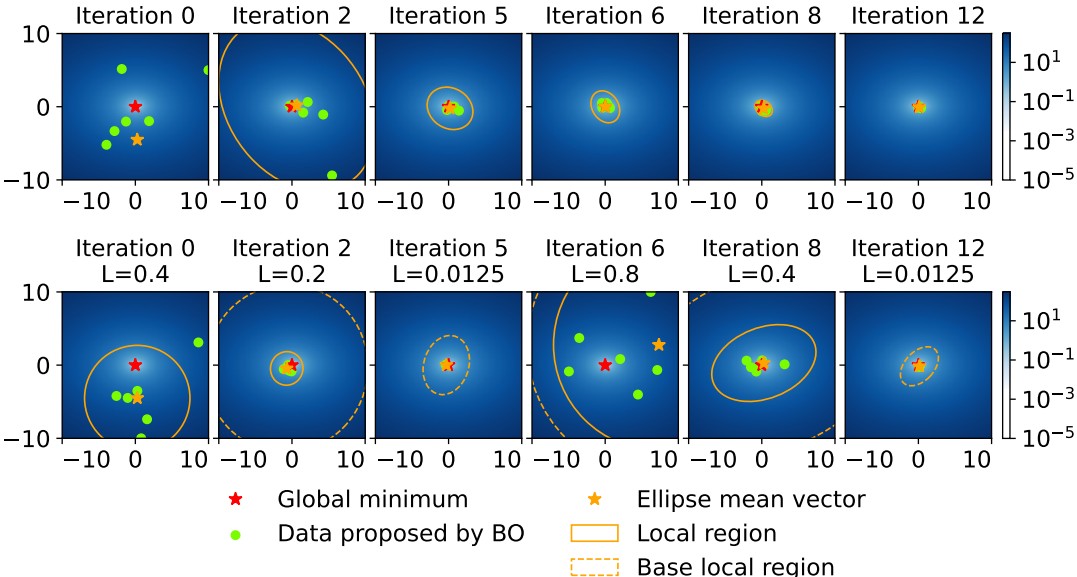

Figure 14: Trajectories of the local regions defined by CMA-based BO methods, `CMA-BO` (upper) and `CMA-TuRBO` (lower), for Ellipsoid 2D function. The function global optimum is at $[0, 0]$.

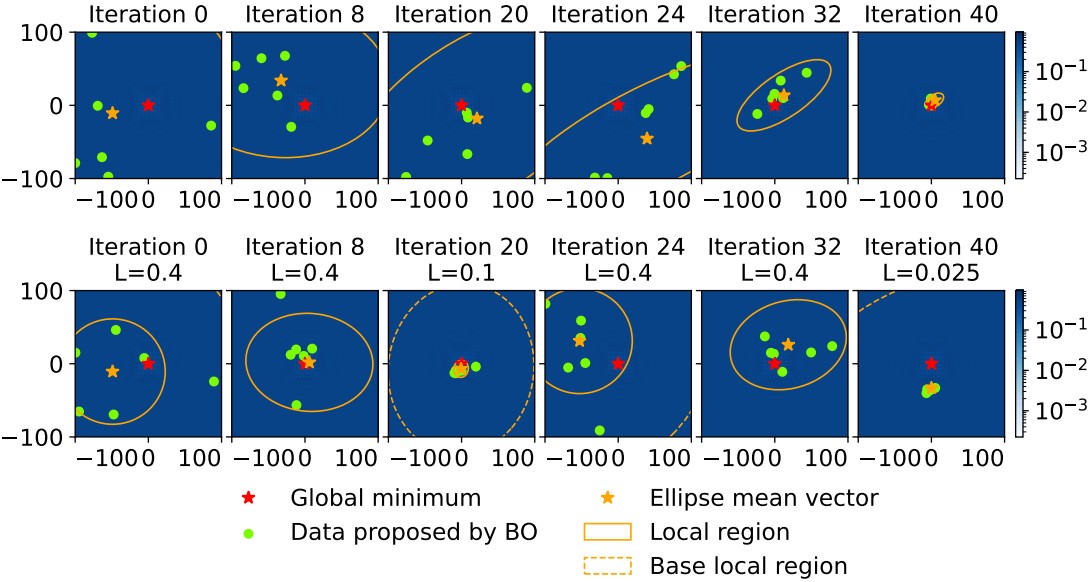

Figure 15: Trajectories of the local regions defined by CMA-based BO methods, `CMA-BO` (upper) and `CMA-TuRBO` (lower), for Schaffer 2D function. The function global optimum is at $[0, 0]$.

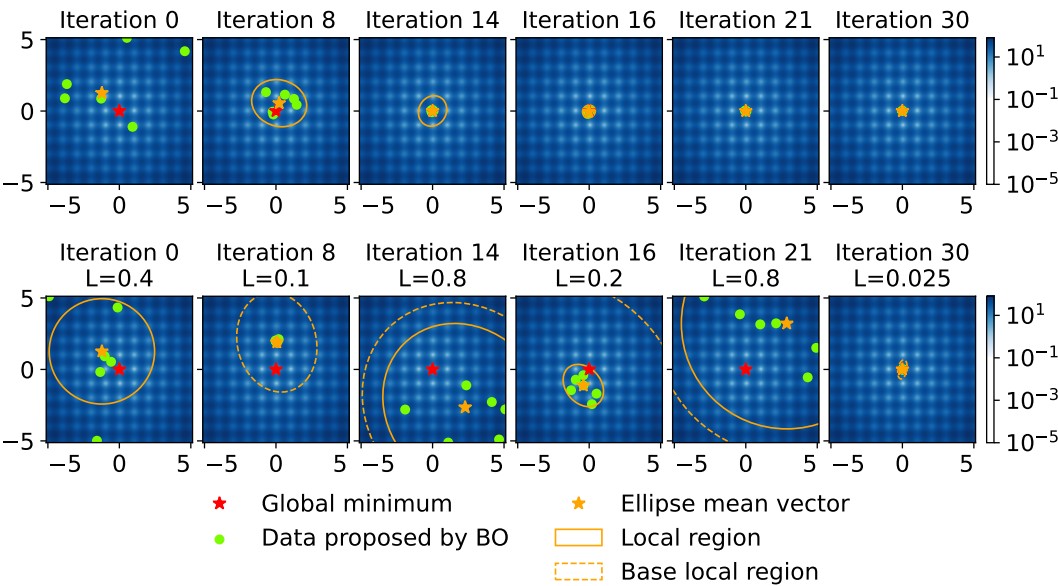

Figure 16: Trajectories of the local regions defined by CMA-based BO methods, `CMA-BO` (upper) and `CMA-TuRBO` (lower), for Rastrigin 2D function. The function global optimum is at $[0, 0]$.

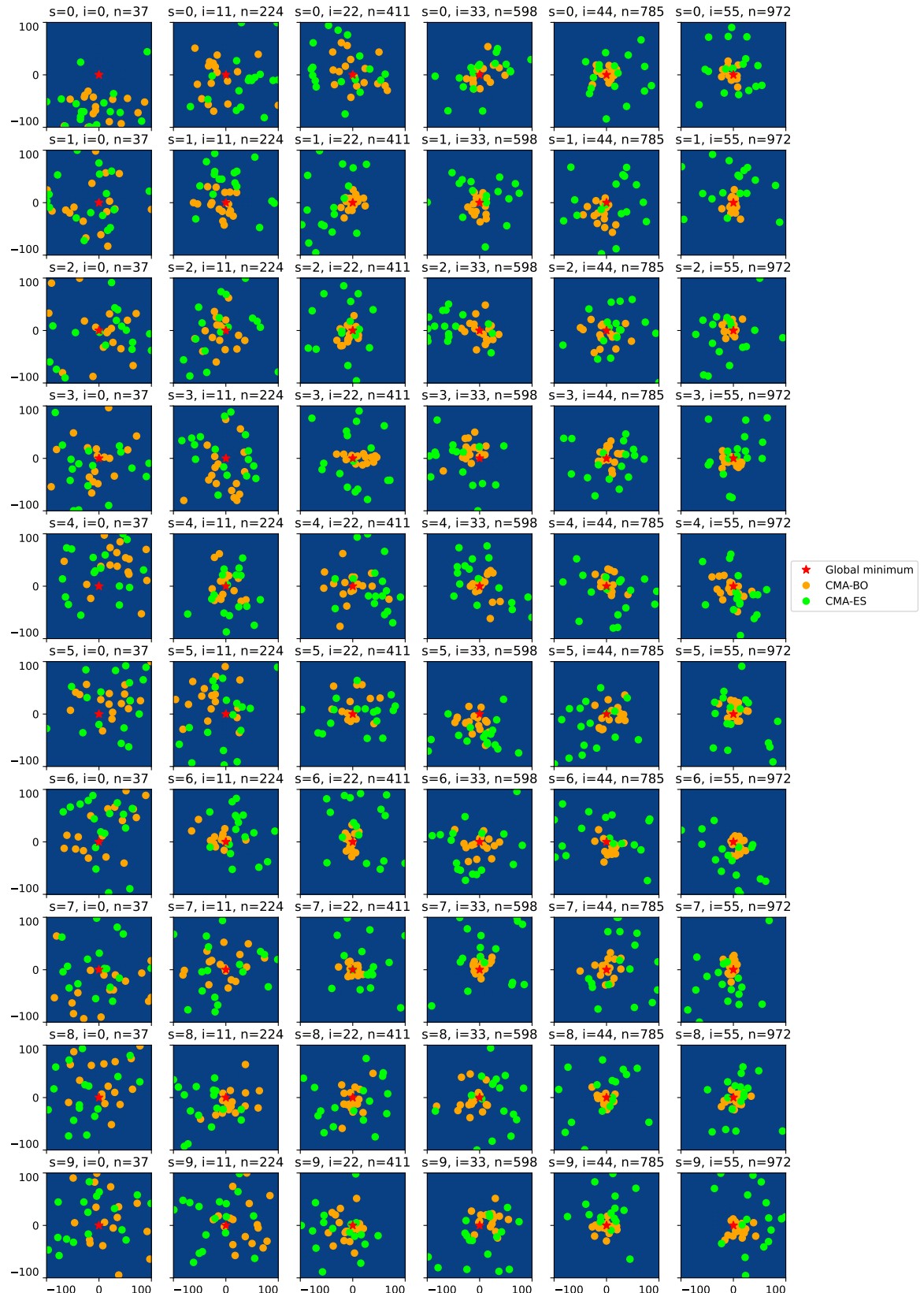

Figure 17: 2D plots for Schaffer2-100D projecting on the 2 effective dimensions for 10 repeats. `CMA-ES` always tends to over explore whilst `CMA-BO` generally finds more meaningful data points.

