# OpenReview forum: "High-dimensional Bayesian Optimization via Covariance Matrix Adaptation Strategy"
_TMLR — Accepted by TMLR_

### Review · Reviewer_pHRT · 2023-11-05

**Summary Of Contributions:**

This manuscript proposes to combine the CMA strategy (from CMA-ES) with that of BO to dynamically adjust trust regions given observed data in order to make high-dimensional BO more tractable. On 5 synthetic objective functions and 3 simulated physics control tasks, they demonstrate that their approach beats many existing baselines.

**Audience:**

Yes

**Claims And Evidence:**

Yes

**Requested Changes:**

- Algorithm 1: It's not super clear what the different roles of Omega and D_lambda are, perhaps that can be clarified in both the pseudo-code and the text.
- The 2 experimental results figures are very cluttered and could probably be refactored into a clearer 3-4 plots with more targeted comparisons (e.g. CMA-XYZ vs XYZ then CMA-XYZ vs MCTS-XYZ, etc).

**Strengths And Weaknesses:**

Strengths:
- Well written text.
- Comparison to an extensive suite of baselines.
- Good coverage of related work with perhaps one exception (Unbounded Bayesian Optimization via regularization).

Weaknesses:
- Minor worry about the sentence: "the rationale behind this step is that in the CMA strategy [...] the observed data points are required to be sampled from the previous search distribution." This argument somewhat breaks when the points are then filtered by Thompson sampling from the GP. The actual distribution from which the points are sampled is a factored distribution where the first is the CMA's normal distribution and the second is p(x* | D) under the GP prior.
- Algorithm 1 could be a bit cleaned up, see next section.
- Experimental results could be a bit cleaned up, see next section.

---

> ### Author Response · Authors · 2023-12-06
> **Response to Reviewer pHRT (part 1)**
>
> Thank you for your insightful and helpful comments. We would like to address your concerns below.
>
> > This argument somewhat breaks when the points are then filtered by Thompson sampling from the GP. The actual distribution from which the points are sampled is a factored distribution where the first is the CMA's normal distribution and the second is p(x* | D) under the GP prior.
> >
>
> We agree that the resulting $\lambda$ data points do not completely follow the CMA normal distribution, and thus, the updates of $m$, $\sigma$, $C$ can cause a bias issue. A potential issue is that the selected data points by BO could be located near a single point or concentrated around a point compared to the original sampling distribution, and thus can cause premature convergence. However, we argue that this bias issue is not significant in our proposed methods which tackle the **high-dimensional BO problem**.
>
> - The first reason is that the high-dimensional setting causes the curse of dimensionality issue. For example, given a 100-dimension search space $[0, r]^{100}$, if we want to create a candidate solution pool where the distance between any two data points is roughly $r/10$ then the number of required data points in the pool is around $10^{100}$, which is prohibitively large. A usual candidate solution pool generally has thousands of data points (in our setting, it is up to 5000 data points), therefore, *the probability for two data points in the candidate solution pool to be very close to each other is low in the high-dimensional setting*.
> - The second reason and also the most important reason is that **BO optimizers always aim to balance exploitation and exploration**. So a BO optimizer will not only pick data points with better function values (*exploitation*), but they will also select data points in regions with high uncertainty in the GP model (*exploration*). Thus, because of exploration, *the $\lambda$  selected candidate solutions by BO are usually not all located near a single point even when the candidate solution pool is large*. Furthermore, in the high-dimensional setting, and with a limited evaluation budget (which is the scenario of BO that tackles expensive black-box functions), the number of observed data points used to build the GP is normally much smaller compared to the search space size, and this results in a GP with high uncertainty in many areas of the search space, hence making BO to propose data points with some levels of randomness.
>
> Therefore, we believe that the bias issue caused by using BO to pick data points to update $m$, $\sigma$, $C$ is not significant in our proposed methods and in our setting. Furthermore, we also would like to argue that in the high-dimensional setting, CMA-ES will suffer the over-exploration issue with the strategy of randomly sampling data points. To validate our arguments on this bias issue, **we added Sec. 5.4.4 to conduct some experiments to analyze this issue**.
>
> - In Fig. 8, we significantly increase the number of data points $n_c$ in the candidate solution pool (double and triple the original amount) and plot the results of our proposed methods. The results show that the performance of our CMA-based BO methods is still equally good or sometimes even better than the results we got with the original setting. This shows that, in the high-dimensional setting with a limited function evaluation budget, *our proposed methods are robust to $n_c$.*
> - In Fig. 9, we show that picking random data points to update $m$, $\sigma$, $C$ as in CMA-ES causes the over-exploration issue in high-dimensional BO. Because of its random sampling strategy, even though it is unbiased, *CMA-ES’s over-explorative behavior can be very costly in the limited budget setting of high-dimensional BO*. For our CMA-based BO methods, BO optimizers can help to select meaningful data points better and thus, help to mitigate the over-exploration issue of CMA-ES.

---

> ### Author Response · Authors · 2023-12-06
> **Response to Reviewer pHRT (part 2)**
>
> **Requested changes**
>
> > Algorithm 1: It's not super clear what the different roles of Omega and D_lambda are, perhaps that can be clarified in both the pseudo-code and the text”
> >
>
> The local dataset $\Omega$ stores the observed data after each iteration and is cleared after the restart strategy is triggered, while $D_\lambda$ simply represents the set of $\lambda$ data points proposed by the BO optimizers. After each iteration, the local dataset is updated by $\Omega \leftarrow \Omega \cup D_\lambda$. We updated Sec. 4.2 and Algorithm 1 to present these ideas better.
>
> > The 2 experimental results figures are very cluttered and could probably be refactored into a clearer 3-4 plots with more targeted comparisons …”
> >
>
> We updated the figures to make them clearer. In Section 5.3 (Experimental results), we divided into 3 categories:
>
> - Comparison against the state-of-the-art BO optimizers.
> - Comparison against the meta-algorithm.
> - Comparison against the related CMA-based methods.
>
> It’s also worth mentioning that we have **added 3 new problems** suggested by reviewer kmik, meaning **now we have evaluated our methods using a total of 11 benchmark problems**, and in our opinion, this is much more extensive than many standard BO papers which usually evaluate around 6-7 problems. Furthermore, we have also **added Secs 5.4.2 and 5.4.3 to analyze the effectiveness of our proposed methods**.
>
> **Finally, we believe we have addressed the reviewer’s concerns. If this is not the case, we hope the reviewer can let us know and we can continue to address the reviewer’s concerns.**

---

### Review · Reviewer_kmik · 2023-11-21

**Summary Of Contributions:**

This paper proposes a novel framework for black-box optimization in high dimensional search space. To address the issue of inefficacy of BO, the proposed approach combines BO and CMA in the following way. The search distribution of CMA, i.e., Gaussian distribution, is maintained and many candidate solutions are generated from the search distribution within 3 sigma region. BO then selects the best lambda candidate solutions to evaluate on the objective. The search distribution of CMA is updated using the selected candidate solutions using the same update formula as the original CMA-ES. Overall, the proposed approach can be recognized as a variant of the CMA-ES with a surrogate model, where the surrogate model is used to select candidate solutions to be evaluated. The main difference between existing surrogate-assisted CMA-ES and the proposed framework is that the surrogate model is used to pre-select the good candidate solutions rather than assigning their artificial objective function values.

The effect of the proposed approach is demonstrated on synthetic problems and three ML-type problems. Different BO approaches with and without the proposed mechanism, as well as the standard CMA-ES and a surrogate-assisted CMA-ES are compared. Compared to the baseline approaches without the proposed mechanisms, the proposed approaches show their superior efficacy with a limited function evaluation budget.

**Audience:**

Yes

**Broader Impact Concerns:**

no concern

**Claims And Evidence:**

No

**Requested Changes:**

Please see the weak points pointed out above.

**Strengths And Weaknesses:**

Strength

The proposed framework is a rather simple framework and can be applied to many BO baselines. The use of the BO in the surrogate-assisted CMA-ES in this paper is different from other surrogate-assisted CMA-ES, hence I see the novelty. (At the same time, I see the weakness  on this point. See below.) The shown empirical performance is promising.

Weakness

1. The expression of the CMA mechanism is buggy. The authors call $\sigma$ the overall standard deviation, but it is indeed the overall variance as the authors define $\Sigma = \sigma C$ rather than the standard notation of $\Sigma = \sigma^2 C$ in the CMA-ES literature. Then, the algorithm description in (4) and below seems wrong. If $\sigma$ is the variance, (4) is fine but the update of $p$ should not be divided by $\sigma$. Otherwise it does not make sense. If $\sigma$ is the standard deviation, (4) is wrong. The second term should be divided by $\sigma^2$ rather than $\sigma$ and the third term should not be divided by $\sigma$. Because of this point, I am not sure how the proposed approach is indeed implemented.

2. The proposed approach breaks one of the main design principle of the CMA-ES, i.e., stationarity. As lambda candidate solutions are selected by BO approaches from the pool of many candidate solutions that are generated from the search distribution, the selected lambda candidate solutions do not follow the Gaussian distribution anymore. Then the update of $m$, $C$, and $\sigma$ is strongly biased. In the situation of the proposed approach, because the best candidate solutions are selected from BO, if the number of points in the candidate solution pool is large, the selected candidate solutions are located near the single point. Even if the number of the points is not so large, the distribution is expected to be concentrated around a point compared to the original sampling distribution (i.e., Gaussian distribution). Then, $\sigma$ and $C$ end up converging at nearly the rate of their learning rate. Such an update is known to tend to cause premature convergence to a not-even-locally-optimal point and hence is avoided in the CMA-ES.

Therefore, I feel that the reported efficacy of the proposed approach is at the cost of the final performance on many multimodal problems. Although the test problems have many local minima, it is not clear whether the better performance of the proposed approach is due to the convergence to a better local minima or just due to the faster convergence to the not-even-locally-optimal point. To evaluate the local search ability and the global search ability, the authors should perform more empirical study on unimodal problems (such as ellipsoid) and multimodal problems (such as rastrigin, shaffer, bohachevsky, etc.). One possibility is to use a standard benchmarking framework, such as BBOB,  to compare the proposed approaches with the baselines.

---

> ### Author Response · Authors · 2023-12-06
> **Response to Reviewer kmik (part 1)**
>
> Thank you for your insightful and helpful comments. We would like to address your concerns below.
>
> > The expression of the CMA mechanism is buggy. The authors call $\sigma$ the overall standard deviation, but it is indeed the overall variance… Then, the algorithm description in (4) and below seems wrong… I am not sure how the proposed approach is indeed implemented.
> >
>
> We indeed had some typos when describing the CMA strategy in Section 4.1. The correct statements are as follows. The notation $\sigma$ corresponds to the overall standard deviation. Therefore, the covariance matrix should be decomposed as $\Sigma=\sigma^2 C$, hence the search distribution in the following sections should be $\mathcal N(m,\sigma^2 C)$. We have updated the texts accordingly in the revised version of our paper. Regarding the description of the algorithm in the paper, we have also adjusted it accordingly. We updated Eq. (4) (second and third terms in the update of C) as described in the CMA-ES algorithm.
>
> Despite the typos, we would like to emphasize that **these typos do not affect the actual implementation of our proposed methods**. In our implementation, we use the “pycma” package (https://github.com/CMA-ES/pycma) from the authors of CMA-ES to implement the CMA strategy. Specifically, whenever we want to update the search distribution $\mathcal N(m,\sigma^2 C)$, we call the update function of “pycma” package (function “es.tell(…)” in line 247 of file “cmabo/cma_bo.py” in our submitted code) and pass the collected $\lambda$ data points. After having the new CMA search distribution, we then continue to formulate our derived local regions for each of the BO optimizers as explained in Sec. 4.2.

---

> ### Author Response · Authors · 2023-12-06
> **Response to Reviewer kmik (part 2)**
>
> > The selected lambda candidate solutions do not follow the Gaussian distribution anymore. Then the update of $m$, $C$, and $\sigma$ is strongly biased. In the situation of the proposed approach, because the best candidate solutions are selected from BO, if the number of points in the candidate solution pool is large, the selected candidate solutions are located near the single point. Even if the number of the points is not so large, the distribution is expected to be concentrated around a point compared to the original sampling distribution. Such an update is known to tend to cause premature convergence to a not-even-locally-optimal point and hence is avoided in the CMA-ES.
> >
>
> We agree that creating a candidate solution pool, and using BO to pick $\lambda$ data points for the updates of $m$, $\sigma$, $C$ can cause a bias issue and yield premature convergence. However, we argue that this bias issue is not significant in our proposed methods which tackle the **high-dimensional BO problems**.
>
> - The first reason is that the high-dimensional setting causes the curse of dimensionality issue. For example, given a 100-dimension search space $[0, r]^{100}$, if we want to create a candidate solution pool where the distance between any two data points is roughly $r/10$ then the number of required data points in the pool is around $10^{100}$, which is prohibitively large. A usual candidate solution pool generally has thousands of data points (in our setting, it is up to 5000 data points), therefore, *the probability for two data points in the candidate solution pool to be very close to each other is low in the high-dimensional setting*.
> - The second reason and also the most important reason is that **BO optimizers always aim to balance exploitation and exploration**. So a BO optimizer will not only pick data points with better function values (*exploitation*), but they will also select data points in regions with high uncertainty in the GP model (*exploration*). Thus, because of exploration, *the $\lambda$  selected candidate solutions by BO are usually not all located near a single point even when the candidate solution pool is large*. Furthermore, in the high-dimensional setting, and with a limited evaluation budget (which is the scenario of BO that tackles expensive black-box functions), the number of observed data points used to build the GP is normally much smaller compared to the search space size, and this results in a GP with high uncertainty in many areas of the search space, hence making BO to propose data points with some levels of randomness.
>
> Therefore, we believe that the bias issue caused by using BO to pick data points to update $m$, $\sigma$, $C$ is not significant in our proposed methods and in our setting.  Furthermore, we also would like to argue that in the high-dimensional setting, CMA-ES will suffer the over-exploration issue with the strategy of randomly sampling data points, making it inefficient (i.e., requires a large number of function evaluations). To validate our arguments, **we added Sec. 5.4.4 to conduct some experiments to analyze this issue**.
>
> - In Fig. 8, we significantly increase the number of data points $n_c$ in the candidate solution pool (double and triple the original amount) and plot the results of our proposed methods. The results show that the performance of our CMA-based BO methods is still equally good or sometimes even better than the results we got with the original setting. This shows that, in the high-dimensional setting with a limited function evaluation budget, *our proposed methods are robust to $n_c$*.
> - In Fig. 9, we show that picking random data points to update $m$, $\sigma$, $C$ as in CMA-ES causes the over-exploration issue in high-dimensional BO. Because of this random sampling strategy, even though it is unbiased, *CMA-ES’s over-explorative behavior can be very costly in the limited budget setting of high-dimensional BO, making CMA-ES inefficient (i.e., requires a large number of function evaluations)*. For our CMA-based BO methods, BO optimizers can help to select meaningful data points better and thus, help to mitigate the over-exploration issue of CMA-ES.

---

> ### Author Response · Authors · 2023-12-06
> **Response to Reviewer kmik (part 3)**
>
> > I feel that the reported efficacy of the proposed approach is at the cost of the final performance on many multimodal problems. Although the test problems have many local minima, it is not clear whether the better performance of the proposed approach is due to the convergence to a better local minima or just due to the faster convergence to the not-even-locally-optimal point. To evaluate the local search ability and the global search ability, the authors should perform more empirical study on unimodal problems (such as ellipsoid) and multimodal problems (such as rastrigin, shaffer, bohachevsky, etc.).
> >
>
> First, we would like to emphasize that in BO, the objective functions are expected to be not only black-box, but also computationally expensive. As a result, the budget for BO problems is often limited. Therefore, a main goal of BO is to optimize the objective function with minimal function evaluations. Majority of state-of-the-art BO optimizers (e.g., TuRBO, BAxUS) aim to find better function values at early iterations. Therefore, in our opinion, it is reasonable to focus on developing a method to be fast in finding good function values at early iterations.
>
> Based on the reviewer’s comments, **we have added new sections (Secs 5.4.2 and 5.4.3) to** **investigate the ability to approach the global optimum of our CMA-based BO methods.** Furthermore, **we also added the results of 3 new problems** requested by the reviewer: Ellipsoid100D, Rastrigin100D, and Schaffer100D.
>
> - In Figs 2, 3, and 4 (Sec. 5.3), we added the results of the 3 new problems. The results show that *our proposed CMA meta-algorithm and the corresponding CMA-based BO methods still outperform existing baselines significantly in these 3 new problems*. In particular, our proposed methods can obtain lower function values than other baselines by a high margin.
> - We added Sec. 5.4.2, which aims to see how our CMA-based meta-algorithm approaches the global optimum compared to the baselines. In Fig. 6, we plot the Euclidean distance between the data points selected by all the methods and the global optimum through iteration for all synthetic problems (4 original problems + 3 newly added problems). Note it’s not possible to plot for real-world problems as we do not know the global optima of these problems. The results show that the *data points selected by our CMA-based methods can come closer to the global optimum than other baselines*. These new results further reinforce the results in Figs 2,3,4 where we show that our proposed methods can obtain lower function values than other baselines.
> - We added Sec. 5.4.3, which aims to evaluate how our CMA-based meta-algorithm locates promising local regions compared to the baselines. In Fig. 7, we plot the Euclidean distance between the centers of the local regions by the methods to the global optimum through iteration for all synthetic problems (4 original problems + 3 newly added problems). We can see that *the local regions defined by our CMA-based methods also come closer to the global optimum compared to the related baselines*. This demonstrates the effectiveness of our proposed CMA-based meta-algorithm in identifying promising local regions.
>
> **Finally, we believe we have addressed all the reviewer’s concerns. If this is not the case, we hope the reviewer can let us know and we can continue to address the reviewer’s concerns.**

---

### Review · Reviewer_6Py4 · 2023-11-24

**Summary Of Contributions:**

The authors consider a meta-algorithm for Bayesian optimization (BO) where the underlying BO model is only fit to subregions. They port an approach from evolutionary algorithms, covariance matrix adaptation (CMA), to control the points that are modeled using the underlying BO method. They show promising results on learning synthetic and "real" functions.

**Audience:**

Yes

**Claims And Evidence:**

Yes

**Requested Changes:**

Edit down to preferably 8, max 10 pages.

**Strengths And Weaknesses:**

The paper is reasonably well written but IMO much longer than it needs to be. There is only one key idea being introduced here, and only really two result figures (figs 2 and 3 are redundant and should be combined), so there is no need for 15 pages. All sections would benefit from aggressive editing, but in particular the repetitive descriptions of how CMA is combined with the different underlying BO methods would be better presented as a succinct table.

These BO methods are in mind a little dissatisfying since they feel a like a collection of heuristics without any theoretical grounding. Indeed, I would have thought CMA ran a risk of being dragged towards wide shallow optima at the expense of deep narrow optima. The empirical results seems strong but it is challenging to assess how well these examples represent the characteristics of real world applications of BO.

Minor comments:
- what is a "heterogenous" objective function? (p1)
- eq 2 is repetitive with the inline eqns after it
- TurBO is introduced twice in different sections
- section 4.1: how is the "true search distribution" defined?

---

> ### Author Response · Authors · 2023-12-06
> **Response to Reviewer 6Py4 (part 1)**
>
> Thank you for your insightful and helpful comments. We would like to address your concerns below.
>
> > There is only one key idea being introduced here, and only really two result figures (figs 2 and 3 are redundant and should be combined), so there is no need for 15 pages. All sections would benefit from aggressive editing, but in particular the repetitive descriptions of how CMA is combined with the different underlying BO methods would be better presented as a succinct table.
> >
>
> We would like to emphasize our contributions to justify the length of our paper.
>
> - Our first major contribution is **to propose a new meta-algorithm based on CMA** to further enhance the performance of existing state-of-the-art (SOTA) BO optimizers.  Based on our experiments, CMA can significantly enhance the performance of existing BO optimizers. Our proposed technique is the kind of technique that can be added to existing BO methods with **minimal overhead** (see Sec. A.8) while **substantially improving their performance and outperforming existing approaches** (see Sec. 5.3). Therefore, we strongly believe this is a significantly novel contribution because developing such a method is not an easy task.
> - Our second major contribution is to derive the corresponding CMA-based BO algorithms for different existing SOTA BO optimizers like TuRBO and BAxUS. **This task is not straightforward, in fact, it is quite challenging due to the different working mechanisms of different BO optimizers**. For example, TuRBO has its own local region adaptation mechanism to shrink or expand, and the shape of the local regions of TuRBO is hyper-rectangle which is very different from our CMA local regions’ shape which is hyper-ellipsoid. So deriving an appropriate technique to embed TuRBO’s strategy into our CMA strategy while enhancing the performance is challenging. For BAxUS, it’s even more challenging as BAxUS is operated within a series of search space projections on different dimensionalities. On top of that, BAxUS also includes the idea from TuRBO within its optimization process. Thus adapting the mechanism of BAxUS to work well with the CMA local region is also a non-trivial task.
>
> Therefore, we strongly believe that the details of the CMA-based BO algorithms obtained when incorporating the CMA strategy with different existing SOTA BO optimizers like TuRBO and BAxUS are worth describing in detail for readers to follow and reproduce our results. We have added some sentences at the beginning of Secs 4.2.2 and 4.2.3 to describe the challenges when deriving these CMA-based BO algorithms.
>
> To address the reviewer’s comments regarding the paper length, we have tried our best to reduce the length of the paper:
>
> - We moved some descriptions of TuRBO (Sec. 2.3), BAxUS (Sec 2.4), the experiment setup (Sec. 5.1), and benchmark problems (Sec. 5.2) to the Appendix.
> - We reduced the sizes of the figures significantly.
>
> Regarding the two results figures (Figs 2 and 3 in the original version), we had both of them to improve the readability and this is indeed the approach used in related work of meta-algorithms [1, 2] when presenting the performance of the algorithms. To further enhance the clarity, we decided to divide all the results into 3 different figures (Figs 2, 3, 4 in the revised version) to justify the performance of the methods in 3 categories: (1) comparison against the BO optimizers, (2) comparison against other meta-algorithms, and (3) comparison against other related CMA-based methods. Note that we‘ve tried to significantly reduce the size of the figures to keep our paper at a reasonable length per the reviewer’s suggestion.

---

> ### Author Response · Authors · 2023-12-06
> **Response to Reviewer 6Py4 (part 2)**
>
> > These BO methods are in mind a little dissatisfying since they feel a like a collection of heuristics without any theoretical grounding.
> >
>
> As discussed in the paper (Section 4.1), **we developed our CMA-based meta-algorithm based on both the theoretical and empirical performance of CMA-ES in working with high-dimensional optimization problems**. Theoretically, it has been shown that the CMA updates can maximize the probability of selecting successful data points (data points with lower function values) during the optimization process. Empirically, it has been shown in many works [3-6] that CMA-ES has a strong performance in optimizing high-dimensional optimization problems.
>
> Regarding the state-of-the-art BO optimizers we incorporate with our CMA-based meta-algorithm like TuRBO and BAxUS, even though they haven’t had full convergence theoretical analysis yet, they have been empirically shown to perform very well in many high-dimensional optimization problems, and are state-of-the-art high-dimensional BO methods [1-4, 7]. That is the main motivation why we use these BO optimizers.
>
> Following the reviewer’s comment regarding the grounding and rationale behind our developed methods, **we have added new sections (Secs 5.4.2, 5.4.3, 5.4.4, A.8, A.10) to extensively analyze the behaviors and effectiveness of our proposed methods**.
>
> We summarize our analysis as follows.
>
> - In Sec. 5.4.2, we plot the Euclidean distance between the selected data points at each iteration and the global optimum of the objective function of all the methods. The results show that the *data points selected by our CMA-based BO methods can come closer to the global optimum than other baselines*. These new results further reinforce the results in Secs. 5.3.1, 5.3.2, and 5.3.3, where we show that our proposed methods can obtain lower function values than other baselines.
> - In Sec. 5.4.3, we plot the Euclidean distance between the centers of the local regions and the global optimum of all the methods through iteration. We can see that the *local regions defined by our CMA-based BO methods come closer to the global optimum than related baselines*.
> - In Secs. 5.4.4 and A.10, we investigate the bias issue of CMA-based BO methods in the high-dimensional setting and the over-exploration issue of CMA-ES. Our experimental results show that the bias issue caused by using BO to select data points for CMA updates is not critical in the high-dimensional setting of BO. Furthermore, our results show that CMA-ES is highly vulnerable to the over-exploration issue in high-dimensional optimization problems, making it inefficient in the limited budget setting - which is the setting of BO. For our CMA-based BO methods, BO optimizers can help to select meaningful data points better, helping to mitigate the over-exploration issue of CMA-ES.
> - In Sec. A.8, we evaluate the running time of our proposed methods. The results show that our proposed methods' running time is similar to the BO optimizers we incorporate within, demonstrating the efficiency of our proposed methods.
>
> > Indeed, I would have thought CMA ran a risk of being dragged towards wide shallow optima at the expense of deep narrow optima.
> >
>
> We kindly ask the reviewer to elaborate more on the reasons behind the statement “… *CMA ran a risk of being dragged towards wide shallow optima at the expense of deep narrow optima*”, so that we could address this point more effectively. As shown in our new analysis in Secs 5.4.2 and 5.4.3, the chosen data points and the local regions defined by our CMA-based BO methods can come much closer to the global optimum than existing baselines.
>
> > The empirical results seem strong but it is challenging to assess how well these examples represent the characteristics of real world applications of BO.
> >
>
> We would like to mention that **we have added the results of 3 new problems** suggested by reviewer kmik. This means that now **we have evaluated our proposed methods on 11 benchmark problems** which are much more extensive than standard BO papers which usually evaluate around 6-7 problems. The benchmark problems we used are extensively used in related works to evaluate high-dimensional BO methods (see Sec. 5.2). We strongly believe the superior performance of our proposed methods on these 11 benchmarks demonstrates the efficacy of our proposed methods.

---

> ### Author Response · Authors · 2023-12-06
> **Response to Reviewer 6Py4 (part 3)**
>
> **Minor comments**
>
> > what is a "heterogenous" objective function?
> >
>
> A heterogeneous objective function is a function that can show different behaviors across the search space [3]. For example, in some parts of the search space, the objective function can be nearly constant, while in other parts it becomes “wiggly”. This makes the fitting of one global surrogate model challenging, as the common surrogate GP often assumes the hyperparameters, such as characteristic lengthscales and the signal variances, to be constant.
>
> > eq 2 is repetitive with the inline eqns after it
> >
>
> The inline equation refers to the special case where GP is used as the surrogate model. We have modified the text to reduce some redundancy.
>
> > TuRBO is introduced twice in different sections
> >
>
> In Sec. 2.3, we introduce TuRBO as it is one of the main BO optimizers we incorporate with our proposed CMA-based meta-algorithm, hence it is required to describe its core idea for deriving CMA-TuRBO later. In Section 3 (Related Works), we briefly mentioned TuRBO as a state-of-the-art method that uses the search space partition approach, which is a foundation for many BO optimizers derived later in the literature. We have moved some descriptions of TuRBO in Sec. 2.3 to the Appendix.
>
> > section 4.1: how is the "true search distribution" defined?
> >
>
> In this sentence “the search distribution […] potentially converge to the true search distribution”, we aim to say that after adequate CMA updates, the search distribution can eventually allocate the highest probability to the global optimum. We have modified the text accordingly.
>
> **Finally, we believe we have addressed all the reviewer’s concerns. If this is not the case, we hope the reviewer can let us know and we can continue to address the reviewer’s concerns.**
>
> References
>
> [1] Wang, L., Fonseca, R., & Tian, Y. (2020). Learning search space partition for black-box optimization using monte carlo tree search. *Advances in Neural Information Processing Systems*, *33*, 19511-19522.
>
> [2] Song, L., Xue, K., Huang, X., & Qian, C. (2022). Monte carlo tree search based variable selection for high dimensional bayesian optimization. *Advances in Neural Information Processing Systems*, *35*, 28488-28501.
>
> [3] Eriksson, D., Pearce, M., Gardner, J., Turner, R. D., & Poloczek, M. (2019). Scalable global optimization via local Bayesian optimization. *Advances in neural information processing systems*, *32*.
>
> [4]  Ben Letham, Roberto Calandra, Akshara Rai, and Eytan Bakshy. Re-examining linear embeddings for highdimensional bayesian optimization. In Advances in Neural Information Processing Systems (NeurIPS), volume 33, pp. 1546–1558, 2020.
>
> [5] Ilya Loshchilov and Frank Hutter. CMA-ES for Hyperparameter Optimization of Deep Neural Networks, 2016.
>
> [6] Masahiro Nomura, Shuhei Watanabe, Youhei Akimoto, Yoshihiko Ozaki, and Masaki Onishi. Warm starting cma-es for hyperparameter optimization. In Proceedings of the AAAI Conference on Artificial Intelligence, volume 35, pp. 9188–9196, 2021.
>
> [7] Papenmeier, L., Nardi, L., & Poloczek, M. (2023). Bounce: a reliable bayesian optimization algorithm for combinatorial and mixed spaces. *arXiv preprint arXiv:2307.00618*.

---

### Author Response · Authors · 2023-12-06
**Overview of our Response**

We thank all the reviewers for their insightful and helpful comments. We would like to summarize the key comments of the reviewers here and highlight the modifications and additional analysis we’ve conducted following the reviewers’ comments.

- Regarding the novelty of our work. We would like to emphasize our major contributions.
    - Our first contribution is to **propose a new meta-algorithm based on CMA** to further enhance the performance of existing state-of-the-art (SOTA) BO optimizers. Our proposed technique is the kind of technique that can be added to existing BO methods with *minimal overhead* (see Sec. A.8) while *substantially improving their performance and outperforming existing approaches* (see Sec. 5.3). We strongly believe this is a significant novel contribution because developing such a method is not an easy task.
    - Our second contribution is to **derive the corresponding CMA-based BO algorithms for different existing SOTA BO optimizers like TuRBO and BAxUS**. Since different optimizers have different working mechanisms, incorporating these optimizers within our proposed meta-algorithm whilst enhancing their performance is not straightforward, and in fact, is a challenging task.
- Regarding the bias issue caused by using BO to pick data points to update the CMA distributions, we argue that this issue is not critical in the high-dimensional BO setting. In fact, the use of BO helps to mitigate the over-exploration issue of CMA-ES in the high-dimensional setting. We have **added Sec. 5.4.4 to analyze this issue**.
- Regarding the grounding and rationale of our proposed methods, we have **added new sections (Secs 5.4.2, 5.4.3, A.8, A.10) to extensively analyze the behaviors, effectiveness, and efficiency of our proposed methods**. Our results reveal that the selected data points by our proposed CMA-based BO methods can come closer to the global optimum than existing baselines. The local regions defined by our methods can also come closer to the global optimum than existing baselines. Finally, the running time of our proposed methods is also similar to the BO optimizers they incorporate within.
- We have **added 3 new problems** suggested by reviewer kmik, and the results show that our proposed methods still outperform baseline methods on all these 3 problems. It’s worth noting that now **we have evaluated our methods using a total of 11 benchmark problems**, and in our opinion, this is much more extensive than many standard BO papers which usually evaluate around 6-7 problems.
- We have put various parts of the paper in the Appendix and also reduced the sizes of the figures to reduce the length of our paper to a reasonable length.

We believe we have addressed all the reviewers’ comments. We have also revised our paper to reflect the new analysis/results and other comments of the reviewers. We strongly believe our paper has enough technical quality, evidence, and merit for TMLR.

---

### Decision · Action_Editor_WUAx · 2023-12-29

**Recommendation:** Accept as is

**Comment:**

This manuscript considers the problem of high-dimensional Bayesian optimization, wherein we seek to optimize an expensive objective function that is typically modeled as a "black-box." This is an important and well-studied problem. The authors attack this problem with a strategy that attempts to partition the high-dimensional domain into smaller local regions that can be explored more effectively. This strategy has the advantage of being compatible with many existing approaches and performs admirably in a series of well-designed experiments.

The reviewers universally agree that the work presented in this manuscript is high quality and of interest to the TMLR audience. They are universal in their recommendation of acceptance.

Throughout the review and discussion period, the reviewers provided feedback and suggestions that the authors have faithfully incorporated into the manuscript. I believe these changes have considerably strengthened the work. I commend both authors and reviewers for their engagement throughout this process.

I believe the manuscript is both suitable for and ready for publication.

**Audience:**

There is no question that the material in this paper would be of interest to a subset of TMLR's audience as it concerns the fundamental and  well-studied problem of high-dimensional Bayesian optimization.

**Claims And Evidence:**

Yes, the claims made in this submission are supported both by clear exposition and convincing experiments.